# Seeing Across Views: Benchmarking Spatial Reasoning of Vision-Language Models in Robotic Scenes

**Zhiyuan Feng**[1][*][†]   **Zhaolu Kang**[2][*]   **Qijie Wang**[1][*]   **Zhiying Du**[3][*][†]
**Jiongrui Yan**[2]   **Shubin Shi**[2]   **Chengbo Yuan**[1]   **Huizhi Liang**[1][†]   **Yu Deng**[4]   **Qixiu Li**[1][†]
**Rushuai Yang**[5][†]   **Arctanx An**[2][†]   **Leqi Zheng**[1]   **Weijie Wang**[6][†]   **Shawn Chen**[6]
**Sicheng Xu**[4]   **Yaobo Liang**[4]   **Jiaolong Yang**[4][‡]   **Baining Guo**[4]

[1]Tsinghua University   [2]Peking University   [3]Fudan University
[4]Microsoft Research Asia
[5]Hong Kong University of Science and Technology   [6]Zhejiang University

🏠 Project Page          Code          🤗 Dataset

## Abstract

Vision-language models (VLMs) are essential to Embodied AI, enabling robots to perceive, reason, and act in complex environments. They also serve as the foundation for the recent Vision-Language-Action (VLA) models. Yet most evaluations of VLMs focus on single-view settings, leaving their ability to integrate multi-view information underexplored. At the same time, multi-camera setups are increasingly standard in robotic platforms, as they provide complementary perspectives to mitigate occlusion and depth ambiguity. Whether VLMs can effectively leverage such multi-view inputs for robotic reasoning therefore remains an open question. To bridge this gap, we introduce **MV-RoboBench**, a benchmark specifically designed to evaluate the multi-view spatial reasoning capabilities of VLMs in robotic manipulation. MV-RoboBench consists of 1.7k manually curated QA items across eight subtasks, divided into two primary categories: spatial understanding and robotic execution. We evaluate a diverse set of existing VLMs, including both open-source and closed-source models, along with enhanced versions incorporating Chain-of-Thought (CoT)-inspired techniques. The results show that state-of-the-art models remain far below human performance, underscoring the substantial challenges VLMs face in multi-view robotic perception. Additionally, our analysis uncovers two key findings: (i) spatial intelligence and robotic task execution are positively correlated in multi-view robotic scenarios; and (ii) strong performance on existing general-purpose single-view spatial understanding benchmarks does not reliably translate to success in the robotic spatial tasks assessed by our benchmark. We release MV-RoboBench as an open resource to foster progress in spatially grounded VLMs and VLAs, providing not only data but also a standardized evaluation protocol for multi-view embodied reasoning.

## 1   Introduction

Vision–language models (VLMs) (OpenAI, 2024; Team et al., 2023; Anthropic, 2024; Zhu et al., 2025; Bai et al., 2025; Liu et al., 2023b) play a pivotal role in Embodied AI, enabling multimodal perception and reasoning for robots while also serving as the foundation for Vision–Language–Action (VLA) models (Zitkovich et al., 2023; O'Neill et al., 2024; Kim et al., 2024;

---

[*]Equal contribution.
[†]Work done during research internship at Microsoft Research.
[‡]Corresponding author.

Table 1: Comparison of spatial reasoning benchmarks. Prior datasets emphasize single-view relations, abstract reasoning, or non-embodied multi-view perception. The "Partial" in "Multi-View" indicates that these datasets contain only a subset of multi-view samples, mixed with single-view inputs. MV-RoboBench uniquely targets **multi-view spatial reasoning within robotic manipulation scenarios**, combining embodiment with multi-view perception.

| Benchmark | Multi-View | Task Category | Environment / Scenario | Annotation | QA |
|---|---|---|---|---|---|
| EmbSpatial-Bench (Du et al., 2024) | ✗ | Spatial | Indoor ScanNet | Template | 3.6K |
| Visual Spatial (Liu et al., 2023a) | ✗ | Spatial | MSCOCO | Template | 10K |
| RoboSpatial (Song et al., 2025a) | ✗ | Spatial | Indoor tabletop | Template | 3M |
| Spatial-MM (Shiri et al., 2024) | ✗ | Spatial | Internet | Template | 2.3K |
| SpatialVLM (Chen et al., 2024) | ✗ | Spatial | WebLi | Template | 546 |
| VSI-Bench (Yang et al., 2025b) | ✗ | Spatial | Indoor egocentric video | Template | 5K |
| OmniSpatial (Jia et al., 2025) | ✗ | Spatial | Internet | Manual | 1.5K |
| ShareRobot (Eval) (Ji et al., 2025) | ✗ | Robotic | Robot manipulation | Manual | 1.2K |
| ERQA (Team et al., 2025a) | Partial | Spatial + Robotic | Human-egocentric + robotic manipulation | Manual | 0.4K |
| MMSI-Bench (Yang et al., 2025c) | Partial | Spatial | Multi-Domain 3D + Egocentric + Driving | Manual | 1.0K |
| All-Angles Bench (Yeh et al., 2025) | ✓ | Spatial | Multi-view photos and videos | Template | 2.1K |
| Ego3D-Bench (Gholami et al., 2025) | ✓ | Spatial | Egocentric 3D navigation | Template | 8.6K |
| **MV-RoboBench (Ours)** | ✓ | **Spatial + Robotic** | **Robot manipulation** | Manual | 1.7K |

Li et al., 2024; Black et al., 2024; Intelligence et al., 2025) that empower robots to operate in complex real-world environments. By leveraging VLMs, VLAs inherit broad multimodal competence while adding the ability to ground decisions in physical execution, positioning them as the backbone of next-generation robotic intelligence.

Unlike generic multimodal reasoning, robots operate in physical environments rather than abstract 2D tasks. Robotic execution naturally requires *spatial intelligence*: the capacity to interpret 3D structure, reason about geometric relationships, and maintain consistency across viewpoints. Single-view inputs are inherently limited by challenges like occlusion, depth ambiguity, and restricted fields of view. *Multi-view observations*, by contrast, offer complementary perspectives that help overcome these limitations. As they become increasingly standard on robotic platforms, multi-view observations enable more robust perception and decision-making.

Although many benchmarks have been proposed to assess the spatial reasoning capabilities of VLMs (Du et al., 2024; Liu et al., 2023a; Shiri et al., 2024; Chen et al., 2024; Song et al., 2025a; Yang et al., 2025b; Jia et al., 2025), they mostly focus on single-view data. ERQA (Team et al., 2025a) contains only a small portion of multi-view data, and the diversity of views remains limited, and the tasks remain relatively basic. Moreover, they often emphasize general spatial intelligence tasks while giving less attention to the embodied, action-oriented requirements of robotic manipulation. ShareRobot (Ji et al., 2025) extends evaluation to embodied robotic tasks but without multi-view perception. MMSI-Bench (Yang et al., 2025c) includes multi-view tasks, but its questions primarily target basic spatial perception and understanding. All-Angles Bench (Yeh et al., 2025) and Ego3D-Bench (Gholami et al., 2025) expose models to multi-view inputs, yet their evaluation remains confined to photographic alignment or navigation-oriented perception rather than embodied multi-view reasoning for manipulation.

To fill this gap, we introduce **MV-RoboBench**, a benchmark specifically designed to evaluate multi-view spatial reasoning in robotic manipulation scenarios. It is built from real robotic demonstrations with synchronized multi-camera views and encompasses both spatial reasoning and robotic execution tasks. The benchmark includes a total of 1.7K carefully-curated QA items by humans, spanning diverse manipulation tasks and environments. It offers a systematic evaluation of whether VLMs can effectively integrate complementary information from multiple camera views to support decision-making for robots in the real world.

Our key contributions are as follows:

- We establish the first benchmark that integrates spatial and robotic reasoning with synchronized multi-view inputs in robotic manipulation scenarios, enabling a thorough evaluation of existing open-source and closed-source VLM models.

- We show through extensive experiments that robotic multi-view scenarios remain significantly challenging. The most powerful VLM models still fall far below human performance

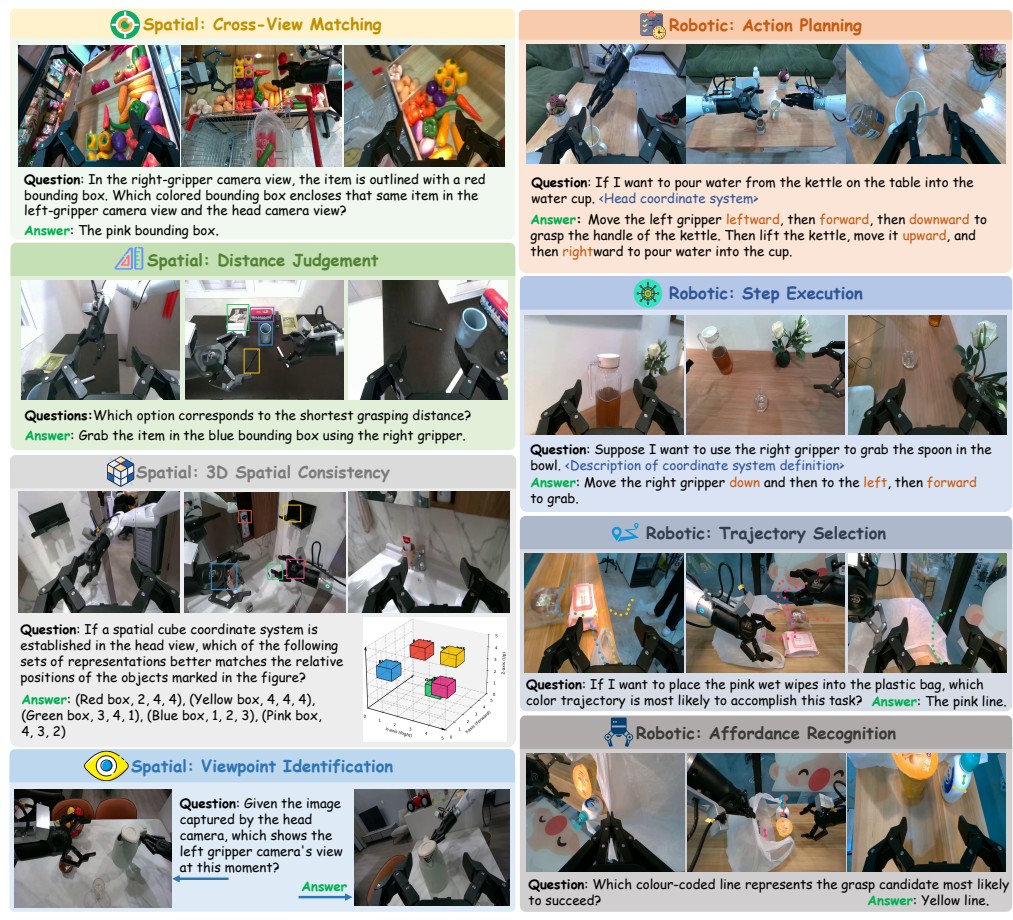

Figure 1: Representative multi-view QA instances from the eight tasks in **MV-RoboBench**, with *spatial* tasks shown on the left and *robotic* tasks on the right. For clarity, only simplified versions with ground-truth answers are presented here, omitting distractors. Full examples are provided in Appendix F.

and many others perform close to random. We further explore CoT-inspired enhancements, which yield mixed and model-dependent effects across models.

- We provide a correlation analysis in multi-view robotic scenarios, uncovering two key findings. First, there is a positive correlation between spatial reasoning and robotic execution. Second, strong performance on general-purpose single-view spatial benchmarks, which assess reasoning from concrete to abstract settings but are devoid of robotic context, does not reliably transfer either to robotic tasks or to spatial reasoning tasks within multi-view robotic scenarios. These findings highlight the unique challenges of multi-view reasoning in robotics and the need for specialized benchmarks like MV-RoboBench.

# 2 MV-ROBOBENCH

## 2.1 OVERVIEW

We introduce **MV-RoboBench**, a benchmark designed to evaluate the multi-view reasoning capabilities of VLMs in robotic manipulation scenarios. It is built from the *AgiWorld* (Bu et al., 2025) and *BridgeV2* (Walke et al., 2023) datasets, spanning both single-arm and dual-arm robotic manipulation settings. In total, we construct 1,708 multiple-choice questions across eight subtasks, each with exactly one correct answer, enabling objective, reproducible, and easily extensible evaluation.

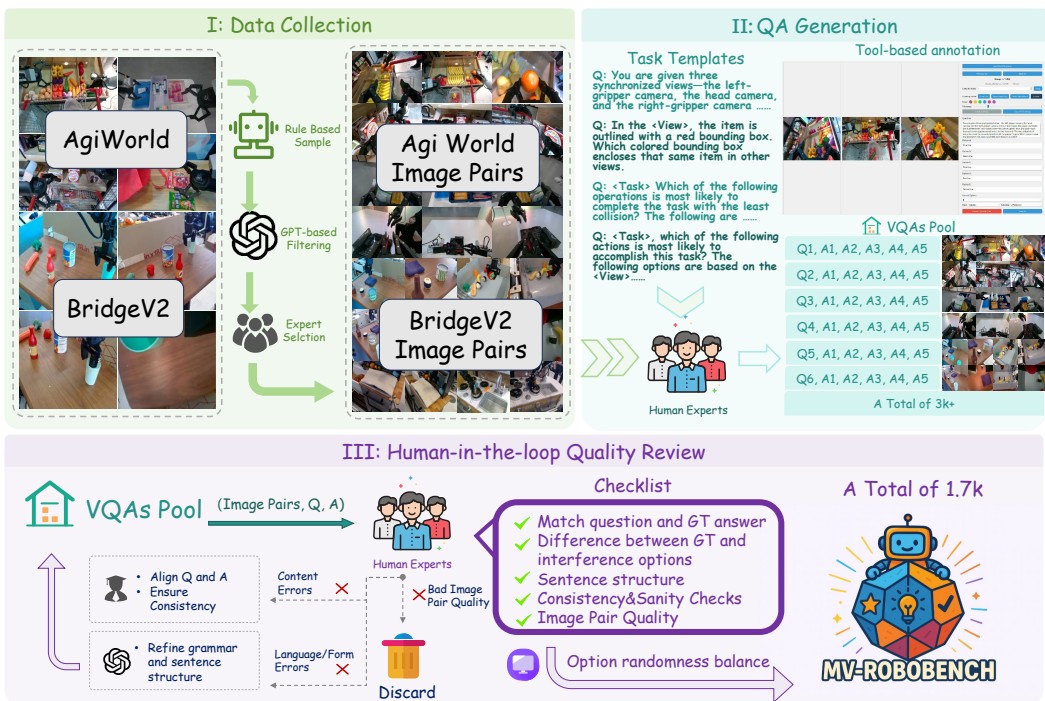

Figure 2: Construction pipeline of MV-RoboBench, consisting of three stages: data collection, QA generation, and human-in-the-loop quality review.

Figure 1 illustrates representative examples from the eight subtasks in MV-RoboBench. To systematically evaluate multi-view reasoning in robotic contexts, we divide the benchmark into two complementary categories: *spatial understanding* and *robotic execution*. Spatial understanding focuses on perception and reasoning across multiple camera views, assessing whether multi-view observations can be integrated into a coherent 3D representation of the scene. Robotic execution, in contrast, extends this spatial reasoning to embodied decision-making, probing whether multi-view information can be effectively leveraged to support planning, execution validation, trajectory feasibility, and affordance reasoning in manipulation tasks.

The four *spatial understanding* subtasks each target a distinct aspect of multi-view perception: *cross-view matching* requires identifying the same object across different viewpoints; *distance judgement* evaluates relative distances between objects; *viewpoint identification* tests the ability to reason about viewpoint transformations; and *3D spatial consistency* probes whether models can maintain consistent relative positions of objects in 3D space. Most of these subtasks rely on paired images as input, emphasizing the integration of complementary viewpoints.

The four *robotic execution* subtasks test whether multi-view information can support embodied decision-making in manipulation. *Action planning* requires choosing an appropriate multi-step sequence to complete a task, while *step execution* focuses on verifying whether the next single-step movement is correct. *Trajectory selection* evaluates the feasibility of candidate motion paths, and *affordance recognition* assesses the feasibility of object-specific interactions. Together, these subtasks emphasize the role of multi-view observations in resolving occlusion and depth ambiguity for embodied decision-making.

## 2.2 BENCHMARK CONSTRUCTION

We design a carefully engineered, multi-stage pipeline that has been iteratively refined to ensure the construction of high-quality QA pairs at scale (Figure 2).

**Data Collection.** We first apply rule-based filtering to synchronized multi-view image pairs to ensure sufficient temporal separation, scene diversity, and visual clarity. GPT-4.1 then serves as an auxiliary filter by checking whether pairs satisfy at least one of the eight task definitions, after which

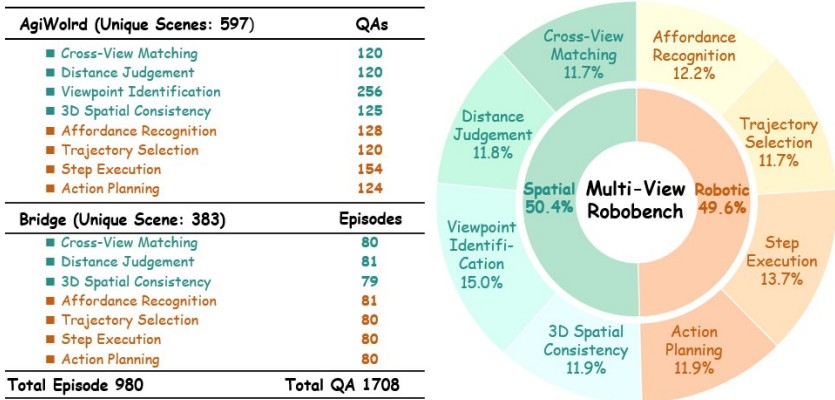

Figure 3: Data distribution of MV-RoboBench, showing QA counts per subtask and dataset source (AgiWorld and BridgeV2), and the overall balance between spatial and robotic domains.

human annotators verify clarity and appropriateness. Importantly, GPT-4.1 is never used to generate QA content but only to assist in candidate triage, and all retained items are manually validated to ensure that genuine multi-view reasoning is required rather than pattern completion.

**QA Generation.** For each subtask, task-specific templates were designed, and trained annotators constructed corresponding five-choice QA pairs from the curated image pairs. During annotation, we explicitly avoided designing overly ambiguous or artificially tricky questions, while ensuring that distractors remain plausible yet clearly distinguishable from the correct option. All annotated items were collected into a shared VQA pool for subsequent refinement. Further implementation details are provided in Appendices E–F.

**Human-in-the-loop Quality Review.** Samples from the VQA pool were iteratively reviewed by trained annotators. Items that did not align with the objectives of the benchmark were discarded, while those with minor issues were revised. Content-related issues were corrected manually to maintain consistency between images and QA, while minor grammar or structural issues were refined with GPT-4.1. The revised items were then returned to the VQA pool for subsequent review and balancing. Accepted items were then rebalanced to randomize answer distributions, ensuring fairness and reducing bias before inclusion in the final benchmark.

Finally, Figure 3 provides a detailed breakdown of MV-RoboBench, showing both per-subtask statistics and the balance between spatial and robotic domains. In addition to the 1,708 QA pairs, the benchmark is derived from 980 episodes, highlighting its grounding in diverse real-world robotic demonstrations.

## 2.3 EXPLORING CoT-INSPIRED ENHANCEMENTS FOR MULTI-VIEW UNDERSTANDING

Recent advances in language reasoning show that Chain-of-Thought (CoT) Wei et al. (2022) prompting can elicit structured intermediate reasoning. This raises the question of whether similar staged reasoning can benefit multi-view understanding in embodied robotic settings, where challenges such as cross-view correspondence, viewpoint alignment under narrow baselines, and consistent geometric fusion persist. Building on this intuition, we explore three CoT-style extensions in the context of multi-view robotic reasoning. First, enriching visual inputs with additional scene descriptions serves as a textual CoT, explicitly verbalizing spatial context that may otherwise remain implicit; to implement this, we adopt GPT-4.1 for generating descriptions. Second, generating additional synthesized viewpoints through novel view synthesis provides a visual CoT, supplying extra visual evidence to support cross-view alignment; to implement this, we adopt VGGT (Wang et al., 2025a)[1] as a representative synthesis baseline. Third, introducing depth priors supplies a structural CoT, adding geometric constraints that reduce ambiguity in 3D reasoning; to implement this, we adopt

---

[1]We also tested several recent novel view-synthesis methods, but they performed poorly in robotic multi-view settings, especially under narrow baselines, cluttered tabletops, and gripper-centric viewpoints.

MoGe-2 (Wang et al., 2025b) for depth estimation. Further implementation details are provided in Appendix C.

## 2.4    FROM PERCEPTION TO ACTION: CORRELATION ANALYSIS

If spatial and robotic reasoning were decoupled, improving view-based perception would not necessarily yield action competence; our correlation analysis directly tests this assumption by leveraging the spatial–robotic task split in MV-RoboBench. We refer to this relationship as the *internal correlation axis*, probing whether stronger spatial perception leads to more reliable robotic execution. Beyond this internal relationship, we define an *external generalization axis* that examines whether spatial intelligence measured in existing single-view benchmarks transfers to embodied multi-view tasks. Unlike single-view settings, which assess perception from a fixed perspective, multi-camera setups demand integrating complementary observations into a coherent 3D understanding. This framing leads to two central questions: (i) how spatial and robotic reasoning relate within multi-view manipulation scenarios, and (ii) whether performance on general single-view benchmarks reliably transfers to multi-view embodied reasoning. We next provide systematic evidence on these issues in Section 4.

Table 2: Evaluation on **MV-RoboBench** under a unified zero-shot prompt. ▮ denotes the best score and ▮ the second-best within each column. Qwen2.5-vl-72B leads among open-source models, while GPT-5 ranks highest overall but still remains far below human accuracy.

| Method | Avg. | Rank | Cross-View Match | Distance Judge | Viewpoint ID | 3D Spatial Consist. | Action Plan. | Step Exec. | Trajectory Sel. | Affordance Rec. |
|---|---|---|---|---|---|---|---|---|---|---|
| | | | **Spatial Tasks** | | | | **Robotic Tasks** | | | |
| **Blind Evaluation** | | | | | | | | | | |
| Random Choice | 19.71 | – | 17.80 | 19.40 | 20.00 | 19.07 | 19.41 | 21.54 | 20.65 | 19.81 |
| GPT-3.5-turbo | 18.52 | – | 15.50 | 22.39 | 20.31 | 12.25 | 21.57 | 18.38 | 23.00 | 16.75 |
| GPT-4-turbo | 22.91 | – | 19.00 | 13.43 | 19.92 | 7.84 | 41.67 | 31.20 | 20.00 | 27.27 |
| **Proprietary Models** | | | | | | | | | | |
| GPT-4o-mini | 22.52 | 8 | 24.00 | 22.89 | 23.44 | 11.76 | 24.51 | 28.21 | 20.50 | 23.44 |
| GPT-4o | 27.59 | 3 | 24.50 | 37.31 | 19.92 | 6.37 | 33.33 | 33.76 | 33.00 | 20.10 |
| GPT-4.1-nano | 20.85 | 9 | 17.50 | 25.37 | 18.75 | 14.71 | 22.55 | 22.22 | 20.00 | 17.22 |
| GPT-4.1-mini | 23.98 | 7 | 28.50 | 33.83 | 25.00 | 7.84 | 26.47 | 21.79 | 32.00 | 18.18 |
| GPT-4.1 | 30.90 | 1 | 26.00 | 43.28 | 32.03 | 6.37 | 29.90 | 31.62 | 41.50 | 28.23 |
| Claude-3.5 | 23.71 | 6 | 17.50 | 27.86 | 20.31 | 8.82 | 34.80 | 20.09 | 33.00 | 27.27 |
| Claude-3.7 | 25.47 | 5 | 18.00 | 35.32 | 20.31 | 6.86 | 36.76 | 29.06 | 34.50 | 22.97 |
| Gemini-2.0-flash | 28.94 | 2 | 28.00 | 32.84 | 21.48 | 7.35 | 32.84 | 29.91 | 52.50 | 20.57 |
| Gemini-2.5-flash | 27.23 | 4 | 26.50 | 37.31 | 27.34 | 6.37 | 34.80 | 30.34 | 42.00 | 19.14 |
| **Proprietary Reasoning Models** | | | | | | | | | | |
| o4-mini | 46.47 | 3 | 21.50 | 48.26 | 26.17 | 65.69 | 74.51 | 63.25 | 44.00 | 25.36 |
| GPT-5-chat | 31.63 | 7 | 30.00 | 42.79 | 31.64 | 4.90 | 36.76 | 40.17 | 38.00 | 27.75 |
| GPT-5-nano | 32.75 | 5 | 21.50 | 33.33 | 17.58 | 56.86 | 39.71 | 35.47 | 31.00 | 26.32 |
| GPT-5-mini | 38.28 | 4 | 22.00 | 49.25 | 25.78 | 72.55 | 66.18 | 48.72 | 47.00 | 27.75 |
| GPT-5 | 56.41 | 1 | 29.00 | 55.22 | 44.14 | 82.35 | 79.41 | 68.38 | 54.50 | 39.23 |
| Claude-3.7-think | 31.67 | 6 | 24.40 | 35.04 | 36.00 | 52.45 | 21.50 | 37.81 | 21.08 | 23.05 |
| Gemini-2.5-pro | 49.52 | 2 | 39.50 | 56.22 | 38.28 | 49.02 | 65.20 | 50.85 | 65.50 | 31.58 |
| **Open-Source Models** | | | | | | | | | | |
| Gemma-3-4b | 19.79 | 11 | 21.00 | 22.89 | 21.09 | 11.76 | 17.65 | 16.67 | 25.50 | 22.01 |
| Gemma-3-12b | 20.49 | 9 | 18.00 | 26.37 | 20.31 | 9.80 | 22.55 | 20.94 | 25.50 | 20.57 |
| Gemma-3-27b | 20.55 | 8 | 21.50 | 23.88 | 20.31 | 9.31 | 20.10 | 23.08 | 29.00 | 17.22 |
| InternVL3-2b | 18.93 | 12 | 16.50 | 15.42 | 20.70 | 20.59 | 17.16 | 20.94 | 21.00 | 19.14 |
| InternVL3-8b | 20.97 | 6 | 19.00 | 21.39 | 26.17 | 12.75 | 26.47 | 21.37 | 20.50 | 20.10 |
| InternVL3-14b | 21.47 | 5 | 19.50 | 22.39 | 24.61 | 10.78 | 23.53 | 23.50 | 24.00 | 23.44 |
| InternVL3-38b | 22.80 | 3 | 24.50 | 25.87 | 23.44 | 6.86 | 27.94 | 25.21 | 27.50 | 21.05 |
| InternVL3-78b | 23.25 | 2 | 19.00 | 28.86 | 23.83 | 11.76 | 29.90 | 29.06 | 26.50 | 21.05 |
| Qwen2.5-vl-3b | 20.37 | 10 | 17.50 | 21.89 | 22.66 | 17.65 | 17.16 | 17.95 | 22.00 | 25.84 |
| Qwen2.5-vl-7b | 20.84 | 7 | 20.50 | 20.40 | 20.70 | 8.82 | 22.55 | 26.07 | 24.50 | 22.49 |
| Qwen2.5-vl-32b | 22.48 | 4 | 20.50 | 25.87 | 25.39 | 10.78 | 24.51 | 19.66 | 30.50 | 22.49 |
| Qwen2.5-vl-72b | 24.29 | 1 | 20.50 | 34.83 | 27.34 | 4.90 | 28.43 | 27.35 | 29.00 | 24.88 |
| **Open-Source MoE Models** | | | | | | | | | | |
| Llama-4-Scout | 22.12 | 2 | 20.50 | 22.39 | 23.83 | 7.35 | 25.49 | 28.21 | 23.00 | 18.18 |
| Llama-4-Maverick | 26.11 | 1 | 14.00 | 42.79 | 17.58 | 5.88 | 37.75 | 37.18 | 36.00 | 20.10 |
| **Human Evaluation** | | | | | | | | | | |
| Human | 91.04 | – | 95.02 | 94.03 | 92.19 | 93.66 | 86.34 | 89.74 | 87.56 | 89.05 |

## 3 EVALUATION ON MV-ROBOBENCH

### 3.1 EVALUATION SETUP

We evaluate a broad spectrum of systems spanning five categories: **Blind Evaluation**, text-only LLMs without visual grounding (Random, GPT-3.5-turbo (Roumeliotis & Tselikas, 2023), GPT-4-turbo (Achiam et al., 2023)); **Proprietary Models**, multimodal systems from major providers, including the GPT-4o family (Hurst et al., 2024), the GPT-4.1 series (OpenAI, 2024), Claude-3.5/3.7 (Anthropic, 2024), and the Gemini-2.x flash family (Team et al., 2023); **Proprietary Reasoning Models**, architectures optimized for multi-step reasoning such as o4-mini (OpenAI, 2025b), the GPT-5 family (chat/mini/nano/full) (OpenAI, 2025a), Claude-3.7-think (Anthropic, 2024), and Gemini-2.5-pro (Team et al., 2023); **Open-Source Models**, community-developed VLMs including the Gemma-3 family (4B–27B) (Team et al., 2025b), the InternVL3 series (2B–78B) (Zhu et al., 2025), and the Qwen2.5-vl series (3B–72B) (Bai et al., 2025); and **Open-Source MoE Models**, namely Llama-4-Scout and Llama-4-Maverick (Meta AI, 2025). Since all tasks are formulated as multiple-choice questions, we adopt answer accuracy as the evaluation metric. This unified format avoids model-specific prompt engineering and ensures a fair cross-model comparison on multi-view reasoning ability. Human evaluations were conducted separately with participants holding a computer science background to serve as a reference point. Further implementation details are provided in Appendix B.

### 3.2 MAIN RESULTS ON MV-ROBOBENCH

Table 2 reveals a consistent trend from perception-oriented systems toward explicitly reasoning-optimized architectures. Proprietary multimodal models such as GPT-4.1 reach 30.90%, while open-source VLMs including Qwen2.5-vl-72B (24.29%) and MoE variants such as Llama-4-Maverick (26.11%) perform moderately lower. The largest gains arise in the proprietary reasoning category: GPT-5 achieves 56.41%, with Gemini-2.5-pro (49.52%) and o4-mini (46.47%) also performing strongly. Figure 4 contrasts leading representative models from each family against human performance, highlighting a substantial remaining gap across both spatial and robotic subtasks.

Task-level analysis shows that **3D Spatial Consistency** is especially challenging. Most non-reasoning models perform near or even below random-choice accuracy (19.07%), indicating that they fail to leverage multi-view information and effectively guess without spatial integration. In contrast, reasoning-enhanced models rise to approximately 49–82%. Robotic subtasks, including Action Planning, Step Execution, Trajectory Selection, and Affordance Recognition, also show substantial improvements under reasoning-based architectures. Planning in particular benefits from richer temporal structure in multi-step options compared to single-step execution evaluation. Human participants nearly solve the benchmark at 91.0%, underscoring both the gains enabled by explicit reasoning and the substantial remaining gap toward human-level multi-view robotic intelligence. Furthermore, we validate the necessity of multi-view inputs in Appendix G, showing that single-view baselines suffer significant performance drops (e.g., ~19% drop in Distance Judgement for GPT-5) due to unresolved depth ambiguities.

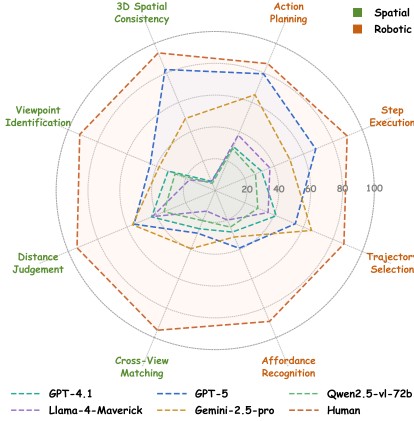

Figure 4: Best-per-group model performance across MV-RoboBench subtasks.

### 3.3 EVALUATION OF CoT-INSPIRED ENHANCEMENTS

As shown in Table 3, CoT-style augmentations exert non-uniform and sometimes counterintuitive effects across models. For Qwen2.5-vl-7B, auxiliary cues bring negligible or even negative changes, with only the depth prior offering a slight gain. Gemma-3-12B, by contrast, benefits substantially from CoT prompting, while textual augmentation and synthetic novel-view generation generally

Table 3: Accuracy of *CoT-style augmentations* on **MV-RoboBench**. $\Delta_s$ and $\Delta_r$ indicate changes on spatial and robotic tasks relative to the origin baseline. Variants: **w cot** = textual prompt, **w text** = descriptive augmentation, **w vggt** = synthetic view, **w depth** = depth prior. ▨ indicates improvement, ▨ degradation.

| | Avg. | Cross-View Match | Distance Judge | Viewpoint ID | 3D Spatial Consist. | $\Delta_s$ | Action Plan. | Step Exec. | Trajectory Sel. | Affordance Rec. | $\Delta_r$ |
|---|---|---|---|---|---|---|---|---|---|---|---|
| Method | | | | Spatial Tasks | | | | | Robotic Tasks | | |
| **Qwen2.5-vl-7b** | | | | | | | | | | | |
| origin | 20.84 | 20.50 | 20.40 | 20.70 | 8.82 | 0.00 | 22.55 | 26.07 | 24.50 | 22.49 | 0.00 |
| w cot | 20.49 (-0.35) | 20.00 | 21.39 | 22.27 | 8.82 | +0.58 | 22.55 | 23.08 | 25.50 | 22.55 | -1.30 |
| w text | 20.90 (+0.06) | 20.00 | 20.40 | 22.27 | 4.41 | -0.70 | 25.98 | 28.21 | 24.50 | 20.10 | +0.82 |
| w vggt | 20.02 (-0.82) | 16.50 | 17.91 | 23.83 | 5.39 | -1.40 | 21.08 | 25.64 | 23.50 | 24.40 | -0.24 |
| w depth | 21.14 (+0.30) | 22.89 | 22.89 | 21.09 | 12.75 | +1.04 | 19.12 | 27.35 | 23.50 | 23.44 | -0.48 |
| **Gemma-3-12B** | | | | | | | | | | | |
| origin | 20.49 | 18.00 | 26.37 | 20.31 | 9.80 | 0.00 | 22.55 | 20.94 | 25.50 | 20.57 | 0.00 |
| w cot | 24.19 (+3.70) | 18.00 | 22.89 | 17.97 | 11.27 | +0.93 | 21.57 | 27.35 | 27.50 | 25.84 | +2.96 |
| w text | 18.43 (-2.06) | 19.00 | 21.89 | 21.09 | 7.84 | -0.94 | 20.10 | 21.79 | 18.50 | 20.10 | -0.47 |
| w vggt | 18.31 (-2.18) | 17.50 | 18.41 | 21.48 | 8.33 | -1.47 | 18.14 | 22.22 | 19.00 | 24.40 | +0.11 |
| w depth | 20.41 (-0.08) | 18.00 | 26.37 | 21.09 | 7.84 | -0.18 | 19.12 | 23.50 | 21.00 | 23.44 | +0.19 |
| **GPT-4.1** | | | | | | | | | | | |
| origin | 29.87 | 26.00 | 43.28 | 32.03 | 6.37 | 0.00 | 29.90 | 31.62 | 41.50 | 28.23 | 0.00 |
| w cot | 29.84 (-0.03) | 28.50 | 40.30 | 29.69 | 6.37 | -1.21 | 28.92 | 30.34 | 46.00 | 22.49 | -0.25 |
| w text | 31.66 (+1.79) | 28.00 | 46.50 | 34.38 | 6.86 | +1.73 | 32.02 | 32.48 | 45.50 | 28.99 | +1.81 |
| w vggt | 28.02 (-1.85) | 29.80 | 38.69 | 31.50 | 4.50 | -1.54 | 29.21 | 31.17 | 40.50 | 27.45 | -1.58 |
| w depth | 33.12 (+3.25) | 30.50 | 45.00 | 34.20 | 10.00 | +3.15 | 31.40 | 33.80 | 47.10 | 28.90 | +2.71 |

degrade performance. GPT-4.1 gains most noticeably from depth priors, with textual augmentation yielding marginal improvements and CoT remaining largely neutral.

Overall, synthetic novel views are more likely to hurt performance, depth priors help only when the backbone has sufficient capacity to exploit geometric cues, and CoT enhancement is most effective for mid-capacity open-source models rather than already over-optimized proprietary ones. These mixed outcomes highlight that multi-view robotic manipulation cannot be reliably improved through generic prompting, suggesting that future progress will require tighter coupling between explicit geometric understanding and structured reasoning rather than shallow prompt-level augmentation. Detailed settings of the three enhancement variants are provided in Appendix C.

## 4 FROM PERCEPTION TO ACTION: CORRELATION AND TRANSFER

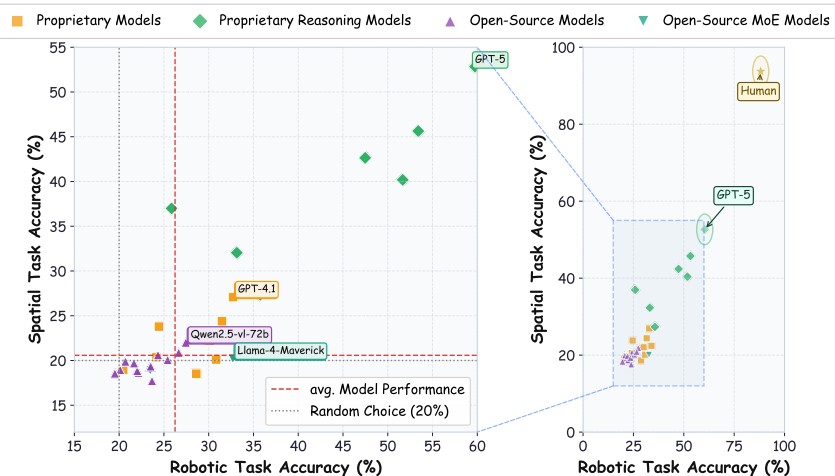

Figure 5: Spatial vs. robotic accuracy on **MV-RoboBench**. Models clustered near the lower-left operate close to random guessing, while reasoning-enhanced proprietary models show a clear upward trend across both axes.

Having established the two analysis axes in Section 2.4—the *internal correlation axis* between spatial and robotic reasoning, and the *external generalization axis* from single-view to multi-view spatial intelligence—we now present empirical evidence along both dimensions.

## 4.1 Internal Correlation: Spatial vs. Robotic Intelligence

As shown in Figure 5, there exists a positive correlation between spatial and robotic accuracy in multi-view manipulation tasks, but this relationship is strongly model-dependent. Proprietary and reasoning-optimized systems exhibit a monotonic trend, where improving spatial perception is accompanied by gains in robotic execution. In contrast, most open-source VLMs cluster near random-choice accuracy, suggesting that without explicit multi-view fusion, perception does not translate into actionable understanding. These results confirm that spatial and robotic reasoning can align, but only when the model possesses sufficient capacity to integrate observations across viewpoints.

## 4.2 External Transferability: Single-View to Multi-View

To assess whether spatial intelligence measured in existing general single-view benchmarks carries over to multi-view robotic manipulation, we use OmniSpatial (Jia et al., 2025) as a reference due to its broad coverage of spatial reasoning. Our reproduced OmniSpatial results are reported in Appendix D.

Figure 6 shows that, aside from proprietary reasoning models, strong single-view accuracy does not reliably transfer to multi-view embodied reasoning. Many models that perform well on OmniSpatial still remain close to random on MV-RoboBench. Even for the highest-performing reasoning models, single-view competence only partially translates, with multi-view accuracy still lagging behind. This indicates that multi-view robotic reasoning introduces fundamentally different demands—particularly on viewpoint integration, occlusion resolution, and spatial fusion—that are not exercised by existing single-view benchmarks, underscoring the necessity of developing dedicated benchmarks tailored for multi-view robotic scenarios.

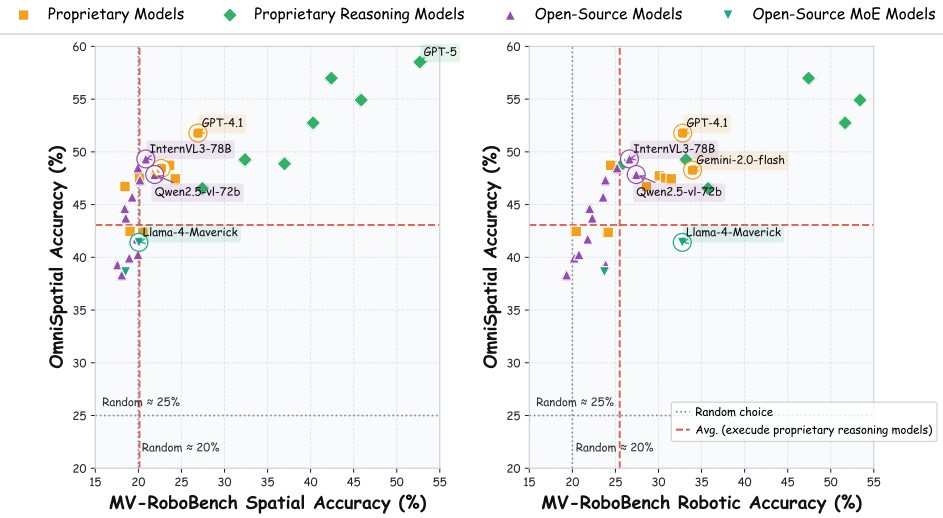

Figure 6: Comparison of model accuracies on OmniSpatial versus MV-RoboBench, with the left plot for spatial subtasks and the right plot for robotic subtasks.

# 5 Ralated Works

## 5.1 Spatial understanding and reasoning in Multimodal LLM

Recent Multimodal Large Language Models (MLLMs) (OpenAI, 2025a; Hurst et al., 2024; OpenAI, 2024; Anthropic, 2024; Team et al., 2023; 2025b; Zhu et al., 2025; Bai et al., 2025; Meta AI,

2025) have demonstrated remarkable progress across diverse tasks, including captioning (Lin et al., 2024; An et al., 2024; 2025), retrieval (Luo et al., 2024; Lin et al., 2025b), planning (Zhou et al., 2024), and even robotic tasks Zitkovich et al. (2023); O'Neill et al. (2024); Kim et al. (2024); Li et al. (2024); Black et al. (2024); Intelligence et al. (2025). However, despite their strong general visual-linguistic competence, these models remain limited in *structured spatial grounding*, particularly when required to maintain 3D consistency, infer depth relationships, or reason across multiple viewpoints (Fu et al., 2024b; Song et al., 2025b; Yang et al., 2025a; Cheng et al., 2024).

To address these challenges, specialized approaches (Cheng et al., 2024; Ma et al., 2025; Zhou et al., 2025; Fan et al., 2025; Liu et al., 2025; Cai et al., 2025; Fu et al., 2024a; Hong et al., 2023; Chen et al., 2024) have attempted to incorporate geometric priors or explicit 3D features into MLLMs. However, such interventions often disrupt pre-trained vision–language alignment, reducing instruction-following robustness. Moreover, even with access to depth or point cloud inputs, current models rarely demonstrate reliable multi-view consistency or explicit exploitation of geometric cues when answering spatial reasoning queries (Zha et al., 2025; Li et al., 2025; Chi et al., 2025). These observations suggest that spatial intelligence in current MLLMs remains predominantly pattern-driven rather than derived from explicit spatial fusion across views.

## 5.2 Benchmarking Spatial and Multi-View Understanding

A growing number of benchmarks have been introduced to evaluate the spatial reasoning abilities of VLMs, as summarized in Table 1. Early efforts such as EmbSpatial-Bench Du et al. (2024), Visual Spatial Liu et al. (2023a), and RoboSpatial Song et al. (2025a) assess template-based object relation reasoning in static single-view scenes. Subsequent datasets, including Spatial-MM Shiri et al. (2024), VSI-Bench Yang et al. (2025b), and SpatialVLM Chen et al. (2024), extend evaluation to egocentric video and free-form spatial queries, but still remain limited to single-view interpretation.

More recent works such as All-Angles Bench Yeh et al. (2025) and Ego3D-Bench Gholami et al. (2025) explicitly evaluate multi-view reasoning, but their tasks are confined to photographic alignment or egocentric navigation perception rather than manipulation-oriented embodied reasoning. By contrast, OmniSpatial Jia et al. (2025) remains a single-view benchmark, although it broadens spatial evaluation to a wider range of reasoning categories. However, all these efforts primarily target general spatial understanding and do not address embodiment or the precision requirements critical for robotic manipulation. In contrast, our **MV-RoboBench** is the first benchmark to couple multi-view spatial reasoning with robotic execution tasks, providing a realistic and comprehensive testbed for embodied multi-view intelligence.

## 6 Discussion and Future Work

Our study highlights three main takeaways. First, multi-view robotic reasoning requires more than perception alone: perception-oriented VLMs yield only modest gains, and only reasoning-augmented systems begin to approach reliable robustness. Second, spatial and robotic intelligence are positively correlated in multi-view manipulation, yet both remain far below human performance, reflecting the absence of robust embodied 3D reasoning. Third, competitive performance on single-view spatial benchmarks does not reliably transfer, revealing a persistent gap between single-view reasoning and embodied multi-view understanding.

Looking forward, progress will likely depend on (i) architectures that explicitly encode geometric priors and enforce cross-view consistency, (ii) training pipelines that align perception with action grounding, and (iii) larger-scale multi-camera datasets that reflect the complexity of real-world manipulation. Our results suggest that scaling perception alone is insufficient—models require explicit reasoning mechanisms to transform multi-view observations into actionable, embodied understanding. By isolating failure modes in multi-view grounding rather than in isolated perception, MV-RoboBench exposes the precise bottlenecks that future embodied AI systems must overcome. We hope it will serve not only as a yardstick but also as a catalyst for developing the next generation of spatially grounded VLMs and VLAs.

ETHICS STATEMENT

This work follows the ICLR Code of Ethics. MV-RoboBench is built entirely from publicly available robotic datasets (AgiWorld and BridgeV2) and does not involve any personally identifiable or sensitive information. All annotations were created by trained annotators under controlled conditions, and we release the benchmark for research purposes only.

REPRODUCIBILITY STATEMENT

We provide detailed descriptions of dataset construction, evaluation setup, and experimental configurations in the main text and appendix. All curated data, task templates, and evaluation code will be released in the supplementary material upon acceptance to ensure reproducibility, while maintaining anonymity during the review process.

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

## A  APPENDIX OVERVIEW

In these supplementary materials, we provide additional details to complement the main paper:

- **Appendix B**: Experimental setup details, including system prompts, inference configurations, and hyperparameter settings for all evaluated models (see Appendix B).

- **Appendix C**: Implementation details of CoT-inspired enhancements, including prompts used for textual augmentation, pipelines for visual augmentation, and configuration for depth priors (see Appendix C).

- **Appendix D**: Complete evaluation setup and results on external benchmarks, covering both *OmniSpatial* and *ERQA* experiments (see Appendix D).

- **Appendix E**: Preparations for building the benchmark, including dataset setup and annotation tool design (see Appendix E).

- **Appendix F**: Detailed process of benchmark construction, including task formulation methodology and annotation workflow (see Appendix F).

- **Appendix G**: Ablation study demonstrating the necessity of multi-view inputs by comparing performance against single-view baselines (see Appendix G).

- **Appendix H**: Analysis of model sensitivity to image orientation (see Appendix H).

- **Appendix I**: Evaluation of model capability to determine "no correct choice" (see Appendix I).

- **Appendix J**: Qualitative error analysis of representative model failures on MV-RoboBench, covering single-view bias, depth and occlusion confusion, frame-of-reference errors, and affordance misunderstandings (see Appendix J).

## B  EXPERIMENTAL SETUP

This appendix provides additional details of the experimental setup used in our evaluation.

### B.1  MODEL ACCESS AND INFERENCE PROTOCOL

All models were evaluated in a *zero-shot* setting with a unified protocol across tasks. Proprietary systems were accessed through their official APIs, while open-source models were run via Hugging-Face implementations.

### B.2  PROMPT TEMPLATES

For reproducibility, we report the exact system- and user-level instructions used across all experiments.

SYSTEM PROMPT

We employed the following JSON-formatted system instruction:

```
1  {
2    "role": "system",
3    "content": "You are an AI assistant performing a harmless academic
         robotics benchmark evaluation. All content is for research purposes.
4
5    You are an evaluator for a robotic vision benchmark.
6    You will be shown a multiple-choice question and a set of candidate
         answers, sometimes with images.
7    Your task is to carefully read the question, consider the provided
         information, and then select the SINGLE best option (A, B, C, D, or
         E).
8
9    Guidelines:
```

```
10    - Always base your answer only on the question and the provided options/
          images.
11    - Do not use external knowledge beyond what is shown.
12    - Output strictly one option letter (A/B/C/D/E).
13    - Do not explain your reasoning unless explicitly requested.
14    - If multiple answers seem plausible, choose the most consistent with
          the given views.
15
16    Answer format:
17    Answer: <option letter>"
18  }
```

### USER PROMPT

Each QA item was wrapped into the following template, where `question` denotes the natural-language question and `opts_str` is the list of candidate options. The corresponding images (base64-encoded) were attached alongside the prompt.

```
1  Question:
2  {question}
3
4  Options:
5  {opts_str}
6
7  Please output a single line of the form:
8  'Answer: X' where X is one of A, B, C, D, E.
```

### B.3   IMAGE ENCODING

All images were provided in base64-encoded format. We followed the OpenAI-style API convention:

```
1  def encode_image_to_base64(image_path: Path) -> str:
2      with open(image_path, "rb") as f:
3          return base64.b64encode(f.read()).decode("utf-8")
```

Encoded images were attached to the user message under the `"image"` field.

### B.4   EVALUATION PROTOCOL

All tasks are framed as multiple-choice QA. Accuracy was computed as the fraction of correctly predicted answers. Each model was evaluated over the entire benchmark without post-hoc filtering. We ensured identical question order and random seeds across runs for fair comparison.

### B.5   HUMAN EVALUATION

We recruited five participants with strong backgrounds in computer science, including PhD, master's, and senior undergraduate students, none of whom were involved in dataset annotation. All participants completed the benchmark under the same interface without access to model outputs. To ensure a fair comparison, we did not impose time limits or prohibit external references, since state-of-the-art models also leverage extensive Internet-scale data. We report the average accuracy of the participants as an approximate human upper bound.

## C   IMPLEMENTATION OF COT-INSPIRED ENHANCEMENTS

This appendix provides implementation details for the three CoT-inspired enhancement strategies explored in Section

### C.1 CHAIN-OF-THOUGHT (COT) PROMPTING

We keep the system prompt unchanged and prepend a single sentence to the *user* prompt:

```
1  You are a careful, step-by-step reasoner. Think concisely.
```

The rest of the user template (question, options, and answer format) remains identical to the zero-shot setting in Appendix B.

### C.2 TEXTUAL AUGMENTATION

To supply richer spatial context, we generated a holistic scene description from the multi-view images using GPT-4.1 (OpenAI, 2024). We prompted the model as:

```
1  These images provide multiple views of the same scene.
2  Based on all of them, provide a single, holistic paragraph
3  describing the entire scene and the spatial relationship
4  between the objects.
```

The generated paragraph was inserted verbatim into the user prompt under a `Context:` header, immediately before the QA item.

### C.3 VISUAL AUGMENTATION VIA NOVEL VIEW SYNTHESIS

To provide cross-view alignment signals, we generated synthetic intermediate views between existing camera perspectives. We experimented with several families of novel view synthesis (NVS) methods:

- **Object-centric synthesis.** Methods such as InstantMesh (Xu et al., 2024) and Trellis (Xiang et al., 2025) are designed for reconstructing individual objects from sparse views. While effective for clean object-level inputs, they proved unsuitable for cluttered robotic scenes, as selecting accurate masks is non-trivial and the outputs often failed to preserve global scene layout (see Appendix Figure 7).

- **Scene-level synthesis.** LVSM (Jin et al., 2024) attempts to interpolate between camera poses with minimal 3D inductive bias. In our robotic setup (e.g., gripper and head-mounted cameras), interpolated views were severely blurred and inconsistent, particularly under narrow baselines and cluttered tabletops (Appendix Figure 8).

- **3D reconstruction-based synthesis.** Geometry-guided methods such as VGGT (Wang et al., 2025a) and $\Pi^3$ (Wang et al., 2025c) leverage explicit multi-view consistency. Compared to generative NVS pipelines, they provided more stable results in cluttered manipulation scenes. Among them, VGGT offered the best trade-off between robustness and efficiency, producing spatially consistent augmentations suitable for our benchmark (Appendix Figure 9).

In practice, we adopted VGGT to generate one interpolated frame between each camera pair, resized to $224 \times 224$, and attached it as an additional input.

### C.4 STRUCTURAL AUGMENTATION VIA DEPTH PRIORS

To provide models with explicit geometric constraints, we augmented each view with predicted depth maps. We considered recent monocular depth estimation approaches, including UniDepthV2 (Piccinelli et al., 2025), but ultimately adopted MoGe-2 (Wang et al., 2025b) as it proved more robust in cluttered indoor manipulation scenes.

For each image, MoGe-2 produced a per-pixel depth map, which was then normalized to a fixed scale range $[0, 1]$ and smoothed with a Gaussian filter to reduce artifacts. During evaluation, depth information was appended as additional input alongside RGB, provided in the following textual form:

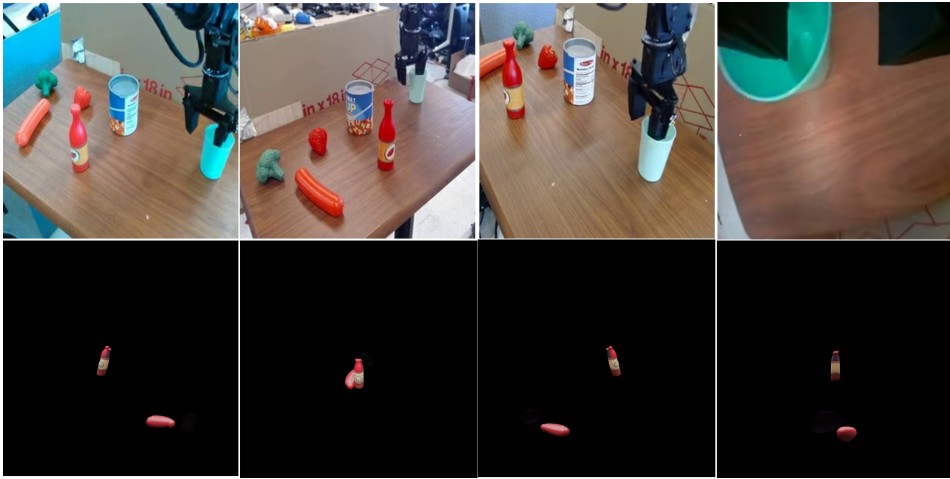

Figure 7: Failure of object-centric synthesis (Trellis). *Top:* original inputs; *Bottom:* synthesized views that fail to capture the full scene.

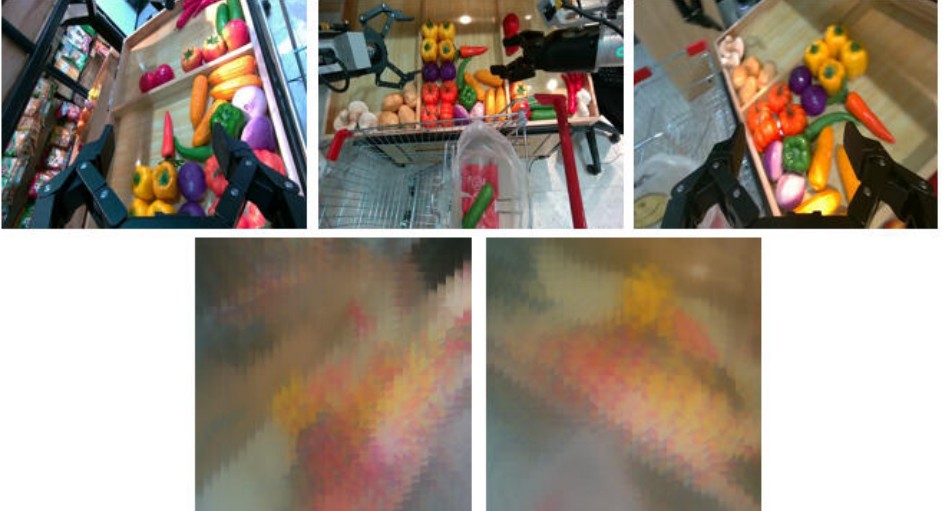

Figure 8: Failure of LVSM scene interpolation. *Top:* original inputs from left gripper, head, and right gripper cameras; *Bottom:* blurry synthesized view from interpolated extrinsics.

```
"text": "Image context: Corresponding estimated depth map.
In this depth map, red areas indicate objects that are closer,
and blue areas indicate objects that are farther away."
```

This additional channel allowed the model to incorporate depth priors when reasoning about occluded or overlapping objects, thereby reducing spatial ambiguity.

## D    EVALUATION ON EXTERNAL SPATIAL BENCHMARKS

Our study focuses on spatial intelligence within robotic operation scenarios. To provide a broader context, we include the **OmniSpatial** benchmark, which spans an unusually comprehensive range of spatial intelligence tasks, from abstract reasoning to concrete domain understanding. We also acknowledge recent efforts such as HSSBench Kang et al. (2025), which benchmarks the capabilities of multimodal models across humanities and social sciences, highlighting the diverse dimensions of evaluation required for MLLMs. Incorporating OmniSpatial allows us to assess whether the

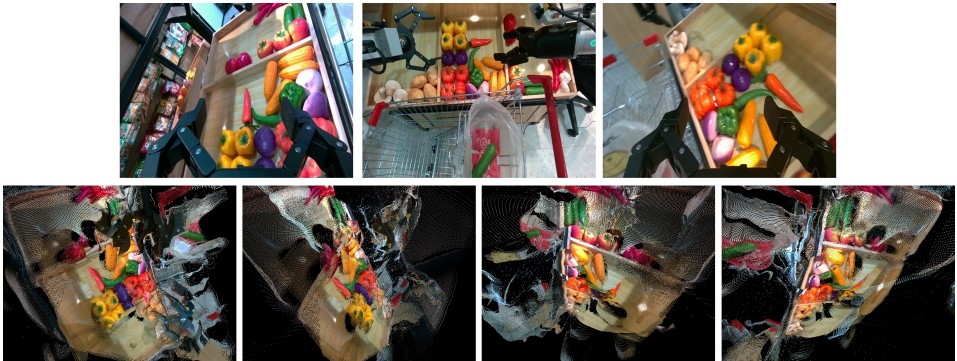

Figure 9: Successful geometry-guided synthesis with VGGT. *Top:* original inputs; *Bottom:* interpolated novel view that preserves object layout and spatial relations.

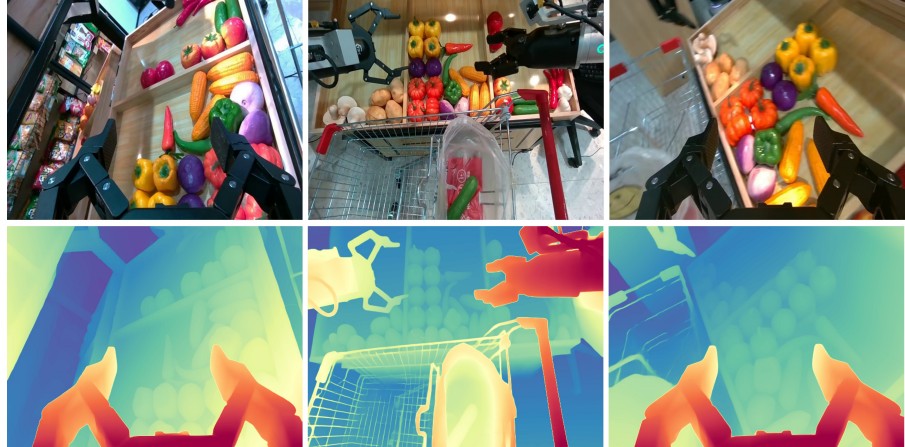

Figure 10: Structural augmentation via depth priors. The top row shows the original RGB images; the bottom row shows the corresponding MoGe-2 depth predictions (red indicates closer, blue indicates farther).

spatial intelligence exhibited by models in general cognitive benchmarks is consistent with their performance in robotics-specific tasks. Table 4 reports these results, where asterisked entries (*) indicate our reproductions, and the remaining scores are taken directly from the OmniSpatial paper to maintain fairness and comparability.

## D.1 ADDITIONAL EVALUATION ON ERQA

Following the reviewers' suggestions, we also evaluated our models on the ERQA benchmark to investigate domain-specific transferability. ERQA focuses on embodied reasoning in simulated environments and includes categories such as Action Reasoning, State Estimation, and a subset of Multi-view Reasoning.

Table 5 reports the detailed performance. While ERQA is domain-relevant, our analysis suggests that it exhibits **low discriminative power** for comparing current SOTA models:

- **Compressed Performance Range:** The overall accuracy gap between smaller open-source models (e.g., Qwen2.5-vl-7b at 43.11%) and SOTA proprietary models (e.g., GPT-4o at 46.00%) is marginally narrow ($< 3\%$).

- **Lack of Gradient:** Most models cluster tightly between 40% and 50% across most subtasks. This "flat" distribution makes it difficult to observe meaningful statistical correlations between model capability and downstream robotic performance.

Table 4: Comparison of model performance on **OmniSpatial**, covering four categories: dynamic reasoning, spatial interaction, complex logic, and perspective taking. Results are reported as average accuracy (%), with asterisked rows (*) denoting our reproduced results.

| Method | Avg. | Dynamic Reasoning | | Spatial Interaction | | | Complex Logic | | Perspective Taking | | |
| | | Manipulation | Motion Analysis | Traffic Analysis | Locali Zation | Geospatial Strategy | Pattern Recog. | Geom. Reasoning | Ego Centric | Allo Centric | Hypothetical |
|---|---|---|---|---|---|---|---|---|---|---|---|
| **Blind Evaluation** | | | | | | | | | | | |
| Random Choice | 24.98 | 24.86 | 26.30 | 25.88 | 23.43 | 27.27 | 21.44 | 24.77 | 22.55 | 24.84 | 25.78 |
| GPT-3.5-turbo | 30.67 | 38.38 | 29.19 | 38.35 | 28.76 | 36.91 | 0.82 | 24.00 | 42.16 | 33.67 | 35.90 |
| GPT-4-turbo | 34.06 | 42.97 | 37.40 | 41.18 | 28.95 | 40.00 | 22.27 | 26.32 | 31.37 | 33.99 | 35.42 |
| **Proprietary Models** | | | | | | | | | | | |
| GPT-4o-mini | 42.64 | 55.95 | 50.29 | 54.59 | 43.43 | 44.91 | 22.47 | 29.42 | 61.57 | 36.76 | 34.22 |
| GPT-4o | 47.81 | 65.54 | 57.23 | 56.47 | 52.38 | 54.09 | 26.29 | 25.48 | 75.98 | 39.49 | 39.76 |
| GPT-4.1-nano | 42.62 | 50.90 | 53.85 | 54.90 | 40.95 | 42.42 | 24.40 | 30.11 | 53.59 | 37.23 | 33.73 |
| GPT-4.1-mini | 48.87 | 64.32 | 56.53 | 59.06 | 60.19 | 56.36 | 29.28 | 30.19 | 72.55 | 39.57 | 39.28 |
| GPT-4.1 | 51.78 | 66.22 | 64.74 | 60.00 | 65.33 | 60.18 | 31.75 | 30.06 | 70.98 | 40.64 | 39.04 |
| Claude-3.5 | 46.86 | 54.05 | 54.57 | 58.12 | 68.38 | 53.09 | 26.60 | 31.74 | 70.00 | 34.79 | 39.52 |
| Claude-3.7 | 47.53 | 57.57 | 55.95 | 56.71 | 63.81 | 59.09 | 29.48 | 28.39 | 72.16 | 36.06 | 36.63 |
| *Gemini-2.0-flash | 48.27 | 62.16 | 55.49 | 50.59 | 60.00 | 54.55 | 22.68 | 34.19 | 74.51 | 39.10 | 45.78 |
| *Gemini-2.5-flash | 47.55 | 67.57 | 52.89 | 63.53 | 55.24 | 57.27 | 29.90 | 23.87 | 79.41 | 36.44 | 44.58 |
| **Proprietary Reasoning Models** | | | | | | | | | | | |
| o4-mini | 52.77 | 72.97 | 59.83 | 60.00 | 73.33 | 61.82 | 34.02 | 36.77 | 73.53 | 40.69 | 40.96 |
| *GPT-5-chat | 46.51 | 59.46 | 46.82 | 56.47 | 59.05 | 53.64 | 34.02 | 25.16 | 70.59 | 41.49 | 45.78 |
| *GPT-5-nano | 49.25 | 63.51 | 58.09 | 51.76 | 65.71 | 50.00 | 32.99 | 26.45 | 70.59 | 42.29 | 42.17 |
| *GPT-5-mini | 57.21 | 74.32 | 61.56 | 67.06 | 79.05 | 72.73 | 35.05 | 36.13 | 81.37 | 47.07 | 46.99 |
| *GPT-5 | 58.51 | 64.86 | 68.79 | 67.06 | 76.19 | 70.00 | 35.05 | 38.06 | 79.41 | 48.94 | 46.99 |
| Claude-3.7-thinking | 48.62 | 57.21 | 59.73 | 53.73 | 67.94 | 57.27 | 30.24 | 28.17 | 68.63 | 37.94 | 36.95 |
| Gemini-2.5-pro | 55.19 | 67.57 | 71.39 | 62.35 | 75.24 | 64.55 | 43.30 | 34.84 | 74.51 | 38.03 | 37.35 |
| **Open-Source Models** | | | | | | | | | | | |
| Gemma-3-4b | 39.79 | 41.89 | 49.71 | 56.47 | 27.62 | 36.36 | 23.71 | 24.52 | 59.80 | 36.17 | 38.55 |
| Gemma-3-12b | 43.71 | 54.05 | 54.91 | 54.12 | 47.62 | 45.45 | 16.49 | 30.32 | 63.73 | 36.70 | 33.73 |
| Gemma-3-27b | 44.75 | 56.76 | 55.78 | 57.65 | 50.48 | 52.73 | 27.84 | 29.03 | 64.71 | 33.51 | 32.53 |
| InternVL3-2B | 37.98 | 50.00 | 40.58 | 43.29 | 40.00 | 40.55 | 21.86 | 28.52 | 55.49 | 35.11 | 33.01 |
| InternVL3-8B | 41.60 | 52.43 | 40.87 | 48.94 | 51.05 | 44.77 | 24.95 | 28.63 | 64.20 | 38.62 | 40.96 |
| InternVL3-14B | 45.94 | 54.32 | 60.17 | 50.35 | 51.81 | 51.45 | 28.04 | 28.26 | 68.04 | 35.37 | 34.46 |
| InternVL3-38B | 48.48 | 63.42 | 63.58 | 54.59 | 58.29 | 50.55 | 29.90 | 28.52 | 72.16 | 36.76 | 33.49 |
| InternVL3-78B | 49.33 | 63.78 | 63.12 | 56.24 | 59.24 | 51.45 | 27.63 | 30.19 | 74.51 | 38.46 | 35.90 |
| Qwen2.5-vl-3b | 40.30 | 55.41 | 47.51 | 46.12 | 42.29 | 44.73 | 32.16 | 23.87 | 59.41 | 33.30 | 30.84 |
| Qwen2.5-vl-7b | 39.18 | 58.38 | 35.09 | 50.12 | 45.33 | 44.00 | 31.13 | 29.42 | 64.51 | 33.19 | 37.35 |
| Qwen2.5-vl-32b | 47.36 | 63.06 | 55.09 | 51.76 | 66.29 | 56.91 | 26.39 | 27.48 | 68.04 | 37.50 | 40.24 |
| Qwen2.5-vl-72b | 47.85 | 58.38 | 60.12 | 50.12 | 59.81 | 53.64 | 26.19 | 33.03 | 71.37 | 36.81 | 36.39 |
| **Open-Source MoE Models** | | | | | | | | | | | |
| *LLama-4-Scout | 38.36 | 51.35 | 39.02 | 51.76 | 34.29 | 42.73 | 20.62 | 22.58 | 52.94 | 39.89 | 34.94 |
| *LLama-4-Maverick | 41.42 | 56.76 | 43.64 | 56.47 | 37.14 | 49.09 | 26.80 | 29.68 | 60.78 | 37.23 | 32.53 |
| **Human Evaluation** | | | | | | | | | | | |
| Human | 92.63 | 96.53 | 97.30 | 92.94 | 97.14 | 94.55 | 91.30 | 87.63 | 99.02 | 95.74 | 93.98 |

Consequently, we retain OmniSpatial as the primary reference for the correlation analysis in the main text (Figure 6), as its wider performance spread provides a clearer signal for measuring general spatial intelligence.

# E PREPARATIONS OF BENCHMARK CONSTRUCTION

## E.1 ANNOTATION TOOL AND INTERFACE

To construct and annotate our dataset, we developed a custom graphical annotation tool based on the `Qt` library, running under the Windows environment. The interface is designed to be clear and lightweight, enabling annotators to efficiently load synchronized multi-view images, draw bounding boxes, trajectories, and affordance lines, and directly export QA items in JSON format that is fully compatible with our evaluation pipeline. Figures 11 illustrate the interfaces used for the AgiWorld and BridgeV2 datasets.

We plan to release this tool as an open-source resource, providing the community with a simple yet powerful interface to facilitate further dataset construction and annotation research.

Table 5: Evaluation results on the ERQA benchmark. Results are reported as accuracy (%). Note the relatively narrow performance gap between open-source and proprietary models compared to MV-RoboBench or OmniSpatial.

| Method | Avg. | Action Reasoning | Multi-view Reasoning | Other | Pointing | Spatial Reasoning | State Estimation | Task Reasoning |
|---|---|---|---|---|---|---|---|---|
| *Proprietary Reasoning Models* | | | | | | | | |
| GPT-5 | **59.34** | 65.71 | 33.33 | 33.33 | 82.35 | 58.33 | 69.81 | 60.53 |
| GPT-5-mini | 54.00 | 54.17 | 32.43 | 42.86 | 55.88 | 58.33 | 61.82 | 65.79 |
| GPT-5-chat | 49.50 | 50.00 | 35.14 | 28.57 | 61.76 | 52.38 | 60.00 | 55.26 |
| GPT-5-nano | 44.00 | 45.83 | 21.62 | 21.43 | 52.94 | 50.00 | 49.09 | 50.00 |
| *Proprietary Models* | | | | | | | | |
| GPT-4.1 | 49.00 | 56.94 | 40.54 | 21.43 | 55.88 | 46.43 | 56.36 | 57.89 |
| GPT-4o | 46.00 | 41.67 | 27.03 | 28.57 | 48.81 | 54.55 | 47.37 | |
| GPT-4.1-mini | 46.00 | 37.50 | 40.54 | 28.57 | 50.00 | 45.24 | 56.36 | 60.53 |
| GPT-4.1-nano | 38.25 | 37.50 | 21.62 | 14.29 | 32.35 | 38.10 | 50.91 | 60.53 |
| *Open-Source Models* | | | | | | | | |
| Qwen2.5-vl-72b | 44.61 | 41.67 | 16.67 | 21.43 | 52.94 | 63.10 | 49.09 | 50.00 |
| Qwen2.5-vl-32b | 44.75 | 38.89 | 32.43 | 21.43 | 58.82 | 52.38 | 54.55 | 47.37 |
| Qwen2.5-vl-7b | 43.11 | 37.50 | 20.00 | 18.18 | 55.88 | 43.37 | 56.36 | 55.56 |

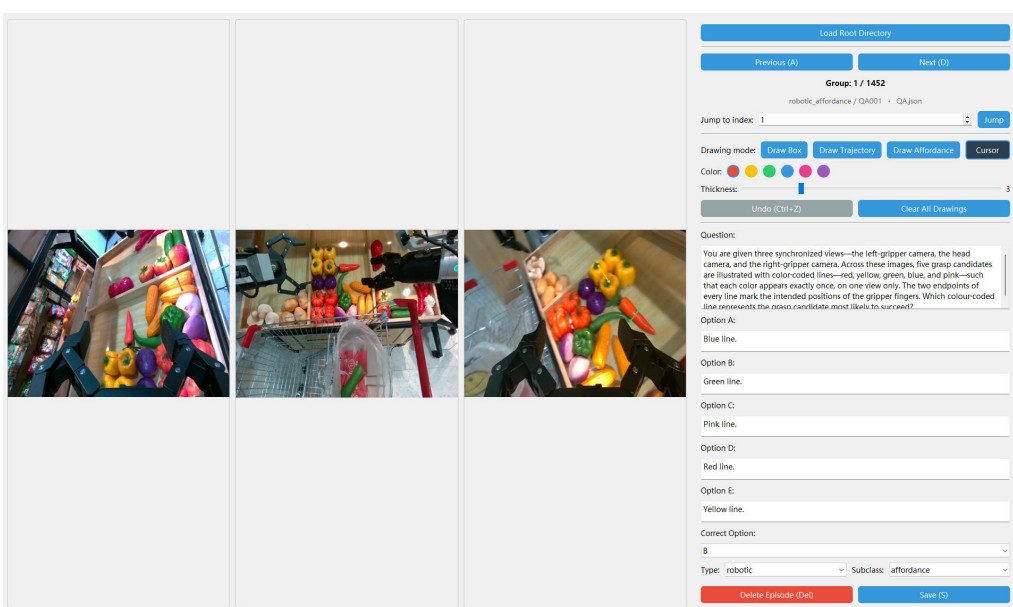

Figure 11: Annotation interface of the AgiWorld label tool, implemented with Qt on Windows. The design emphasizes clarity and ease of use for multi-view annotation.

### E.2 PRE-GENERATION OF IMAGE PAIRS

Before QA construction, we first pre-generated candidate image pairs from both datasets. For the *AgiWorld* dataset, we randomly sampled image pairs with the constraint that the interval between two selected frames was at least ten frames. For the *BridgeV2* dataset, we only considered videos with four available perspectives and similarly enforced a minimum interval of ten frames between sampled images. To ensure diversity, sampling was performed as evenly as possible across videos and tasks.

After this automatic step, each image pair was manually inspected by human annotators, and only those judged suitable for QA were retained. At this stage, we obtained more than 3,000 high-quality image pairs, which served as the foundation for constructing the benchmark. The perspective identification task required a different setup, and its details are described separately in Appendix C.

### E.3 DEFINITION OF THE COORDINATE SYSTEM

To ensure a consistent interpretation of spatial relations across different camera views, we define a standardized right-handed orthogonal coordinate system tied to each camera frame. The construction proceeds as follows:

1. $z$-**axis (vertical).** Let $\mathbf{g}$ denote the gravity vector, pointing downward. We define

$$\hat{\mathbf{z}} = -\frac{\mathbf{g}}{\|\mathbf{g}\|},$$

so that the $+z$ direction points upward (opposite to gravity) and $-z$ points downward.

2. $y$-**axis (forward/backward).** Let $\mathbf{c}$ denote the camera optical axis. Project $\mathbf{c}$ onto the plane orthogonal to $\hat{\mathbf{z}}$:

$$\mathbf{c}_\perp = \mathbf{c} - (\mathbf{c} \cdot \hat{\mathbf{z}})\hat{\mathbf{z}}.$$

Normalizing gives

$$\hat{\mathbf{y}} = \frac{\mathbf{c}_\perp}{\|\mathbf{c}_\perp\|},$$

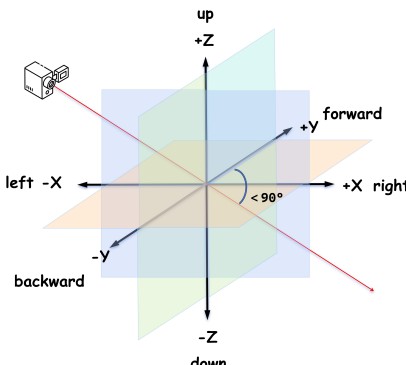

Figure 12: Illustration of the right-handed coordinate system defined relative to each camera.

with orientation chosen so that the angle between $\hat{\mathbf{y}}$ and $\mathbf{c}$ is strictly less than $90°$. By convention, $+y$ corresponds to *forward*, while $-y$ corresponds to *backward*.

3. $x$-**axis (left/right).** Finally, the $x$-axis is determined by the right-hand rule:

$$\hat{\mathbf{x}} = \hat{\mathbf{y}} \times \hat{\mathbf{z}}.$$

This ensures $+x$ points to the right side of the camera's perspective and $-x$ to the left.

**Directional convention.** In summary, $+z$ = upward, $-z$ = downward; $+y$ = forward, $-y$ = backward; $+x$ = right, $-x$ = left. Figure 12 provides an illustration of this definition.

### E.4 TOOL FOR SPATIAL CUBE REASONING

To construct the spatial cube reasoning task, we developed an interactive visualization tool that renders a standardized $5 \times 5 \times 5$ cube grid aligned with the camera coordinate system, where the $x$-, $y$-, and $z$-axes correspond to the *right*, *forward*, and *up* directions. Annotators can place colored unit cubes at integer grid coordinates, assign labels, and interactively edit or regenerate cube configurations.

This design enables rapid prototyping of spatial arrangements and provides a consistent interface for generating QA items that require reasoning about relative positions and geometric relationships in 3D space. The tool also supports keyboard-based coordinate input for efficient and reproducible annotation.

### E.5 HUMAN ANNOTATION PROTOCOL AND QUALITY ASSURANCE

To ensure transparency in benchmark construction, we provide detailed information about the annotators, training procedures, human effort, and the quality-control pipeline adopted throughout the multi-stage annotation process.

#### E.5.1 ANNOTATOR TRAINING AND TASK UNDERSTANDING

All annotators participating in the construction of MV-RoboBench were senior undergraduate students or Ph.D. candidates in computer science or closely related fields. Before large-scale annotation began, we conducted a structured multi-stage training process to ensure that annotators had a rigorous understanding of the purpose and design philosophy of each subtask. First, for each subtask, we provided a conceptual overview explaining *why* the subtask was designed and *what specific*

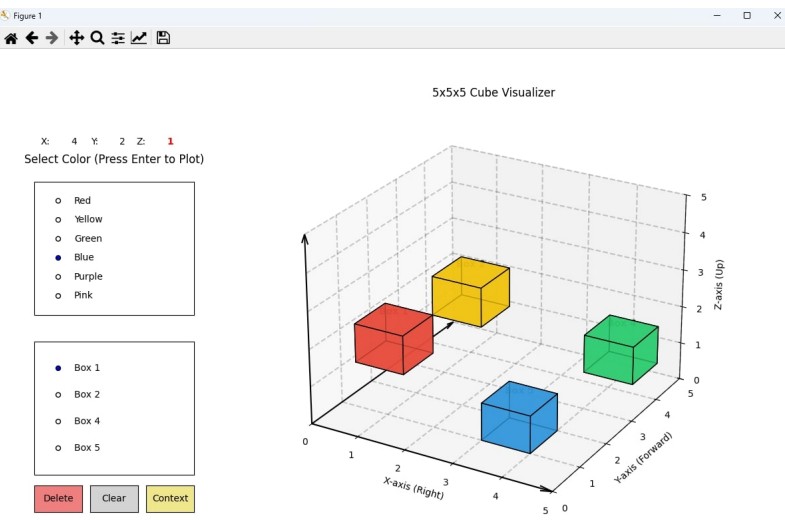

Figure 13: Screenshot of the spatial cube reasoning tool. Annotators can add, label, and manipulate colored cubes within a standardized $5 \times 5 \times 5$ grid to construct 3D reasoning problems.

*aspect of multi-view robotic reasoning it aims to evaluate*. For example, some subtasks target cross-view correspondence, others highlight 3D spatial understanding, action feasibility, or multi-step execution consistency. This high-level motivation helped annotators internalize the intended reasoning challenge behind each category rather than focusing solely on the mechanics of QA creation. Second, we supplied a curated set of high-quality QA examples for each subtask, including both well-constructed samples and typical failure cases. These examples illustrated desirable properties such as clear problem formulation, meaningful distractor design, and unambiguous ground-truth answers. Annotators were instructed to study these examples closely to understand how a robust QA item should be structured. Finally, annotators completed a trial stage in which they produced small batches of QA items. All trial results were reviewed individually by the authors, and detailed feedback was provided for every ambiguous, incorrect, or poorly structured item. Annotators revised their samples accordingly, and only after completing this iterative refinement stage were they allowed to contribute to the full annotation pipeline.

### E.5.2 HUMAN EFFORT ESTIMATION

The construction of MV-RoboBench required substantial human effort across several stages of data preparation, annotation, and verification. We provide an estimate of the total annotation effort below. **Collection and Filtering of Image Pairs (∼200 hours).** This stage involved selecting suitable datasets, writing scripts to automatically pre-filter candidate image pairs, and manually examining the automatically retrieved pairs to determine whether they exhibited clear multi-view correspondences appropriate for downstream QA construction. Annotators carefully removed pairs containing occlusions, poor synchronization, or ambiguous spatial relationships to ensure that only high-quality candidates entered the QA generation stage. **QA Construction and Iterative Refinement (∼600 hours).** This stage accounted for the largest portion of the human effort. The workload included multiple rounds of internal discussion to finalize subtask definitions and question formats, training annotators on the annotation protocol, and iterative communication between authors and annotators to refine how each subtask should be expressed. The actual annotation time—spanning bounding-box drawing, spatial-cube configuration, distractor design, and multi-view reasoning checks—was intentionally left flexible to allow annotators to focus on ensuring that each QA item was both correct and diverse. **Cross-Checking and Validation (∼400 hours).** After QA items were generated, multiple annotators independently reviewed all samples to identify ambiguous phrasing, weak distractors, or incorrect reasoning chains. Items flagged during this stage were either revised through further discussion or discarded entirely. This iterative multi-annotator cross-validation stage was critical for improving the reliability and robustness of the final benchmark.

### E.5.3 QUALITY-CONTROL PROCEDURES

Our benchmark follows the construction flow illustrated in Figure 2, but in this section we focus specifically on the quality-control mechanisms incorporated into each stage rather than the pipeline itself. The goal is to ensure that every QA item in MV-RoboBench is unambiguous, visually grounded, and aligned with the intended reasoning challenge of its corresponding subtask. **Initial Image-Pair Screening.** Before any annotation begins, we employ a two-stage filtering process to guarantee that only high-quality visual inputs enter the QA construction pipeline. First, we use GPT-based filtering to select image-pair candidates that satisfy the definition of each subtask. Second, trained annotators manually verify these candidates by checking whether the images exhibit stable multi-view correspondences, sufficient visual clarity, and the absence of severe motion blur or occlusion. Only pairs judged to be suitable for at least one subtask proceed to the annotation stage. **Annotation with Structured Distractor Design.** After subtask definitions are finalized, annotators—who have undergone dedicated training (Section E.5.1)—construct QA items following standardized guidelines. A critical aspect of our quality control is the design of distractors: each question contains one correct answer and four distractors, among which annotators deliberately create one or two *hard distractors* that closely resemble the correct answer, while the remaining distractors are intentionally more distinct. This structure ensures both a meaningful level of difficulty and a clear separation between high-level reasoning errors and trivial misunderstandings. During annotation, annotators also verify that the correct answer is uniquely supported by the visual evidence and that no distractor inadvertently becomes correct under alternative interpretations. **Multi-Annotator Verification and Iterative Revision.** Once QA items are created, they are added to a shared VQA pool and undergo multi-round cross-checking. Multiple annotators independently review each sample. If *any* reviewer finds a QA item ambiguous, poorly structured, or misaligned with the intended subtask, the item is immediately flagged. Flagged items are either revised through further discussion—during which annotators and authors jointly inspect the visual evidence and reasoning steps—or discarded entirely. Revised items re-enter the VQA pool for additional rounds of validation. This iterative process continues until all items satisfy strict correctness, clarity, and reasoning requirements. Through this combination of automated pre-filtering, human verification, structured annotation protocols, and multi-round cross-validation, MV-RoboBench achieves a high degree of reliability and robustness across all subtasks.

## F DETAILS OF BENCHMARK CONSTRUCTION

In this appendix, we describe the construction details of each subtask included in our benchmark. As introduced in Appendix E.2, we first obtained a large collection of high-quality image pairs from AgiWorld and BridgeV2 through automatic sampling and manual filtering. These image pairs serve as the common starting point for constructing the majority of subtasks, while the perspective identification task required a different setup and is discussed separately later in this section.

For clarity, we organize this appendix by task category. We first present the four **spatial** subtasks, which focus on multi-view scene understanding: Cross-View Object Matching, Distance Judgement, Viewpoint Identification, and 3D Spatial Consistency. We then describe the four **robotic** subtasks, which extend spatial reasoning to manipulation scenarios: Action Planning, Step Execution, Trajectory Selection, and Affordance Recognition. Finally, we conclude with a summary that highlights the complementarity of these subtasks and provides an overview table (Table 6).

### F.1 CROSS-VIEW OBJECT MATCHING

This subtask belongs to the **spatial** category and evaluates whether a model can recognize the same object across different camera viewpoints. In the construction process, one reference view is selected, where the target object is highlighted with a red bounding box. In the remaining synchronized views, candidate objects are marked with bounding boxes of different colors. The model is then asked to identify which candidate corresponds to the same object as the red box in the reference view.

To avoid trivial solutions based only on object category or color cues, distractor candidates are carefully chosen to be visually plausible. These include objects of the same category, those in

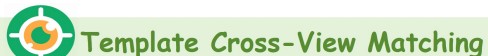

**Template Cross-View Matching**

In the right-gripper camera view, the item is outlined with a red bounding box. Which colored bounding box encloses that same item in the left-gripper camera view and the head camera view?

Rules
- You can replace the orange nouns with labeled objects, or you can just use item instead.
- You can replace the right-gripper with left-gripper or head
- The blue text is a template description, which can be copied directly.

**Sample of Cross-View Matching**

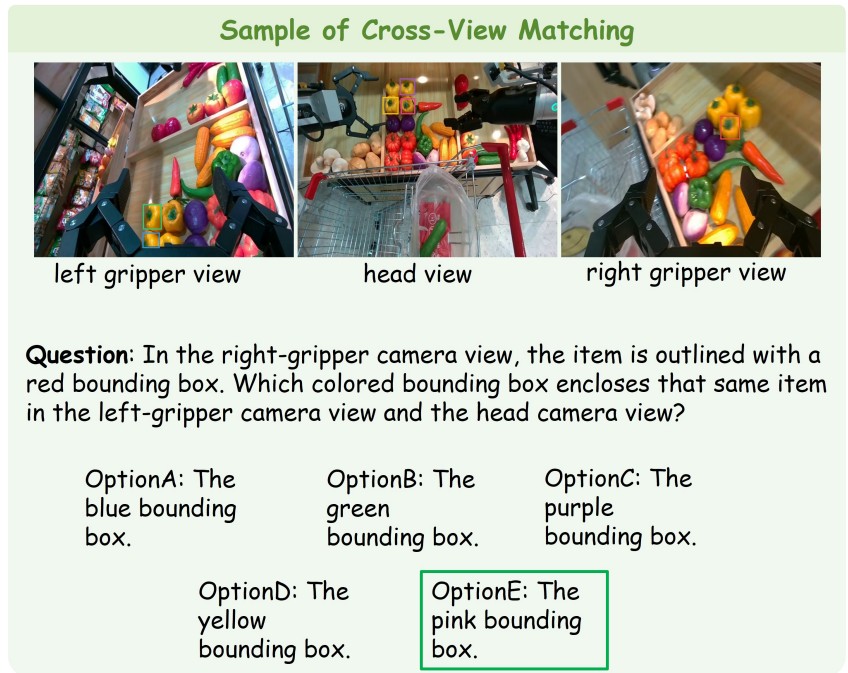

left gripper view          head view          right gripper view

**Question**: In the right-gripper camera view, the item is outlined with a red bounding box. Which colored bounding box encloses that same item in the left-gripper camera view and the head camera view?

OptionA: The blue bounding box.

OptionB: The green bounding box.

OptionC: The purple bounding box.

OptionD: The yellow bounding box.

OptionE: The pink bounding box.

Figure 14: Example of *Cross-View Object Matching* constructed from the AgiWorld dataset. The reference view marks the target with a red bounding box; other views contain color-coded candidate boxes, one of which corresponds to the ground-truth object.

close proximity, or partially overlapping instances, making the task a genuine test of cross-view association.

Figures 14 and 15 show representative examples of this subtask, constructed from the AgiWorld and BridgeV2 datasets, respectively.

## F.2    DISTANCE JUDGEMENT

This subtask belongs to the **spatial** category and evaluates a model's ability to reason about relative distances using synchronized multi-view observations. In each problem, one selected view presents several candidate objects, each marked with a colored bounding box. The model is asked to determine which candidate corresponds to the shortest (or, alternatively, the longest) grasping distance relative to the specified gripper. Other synchronized views provide additional context, requiring the model to integrate information across perspectives to resolve depth ambiguities.

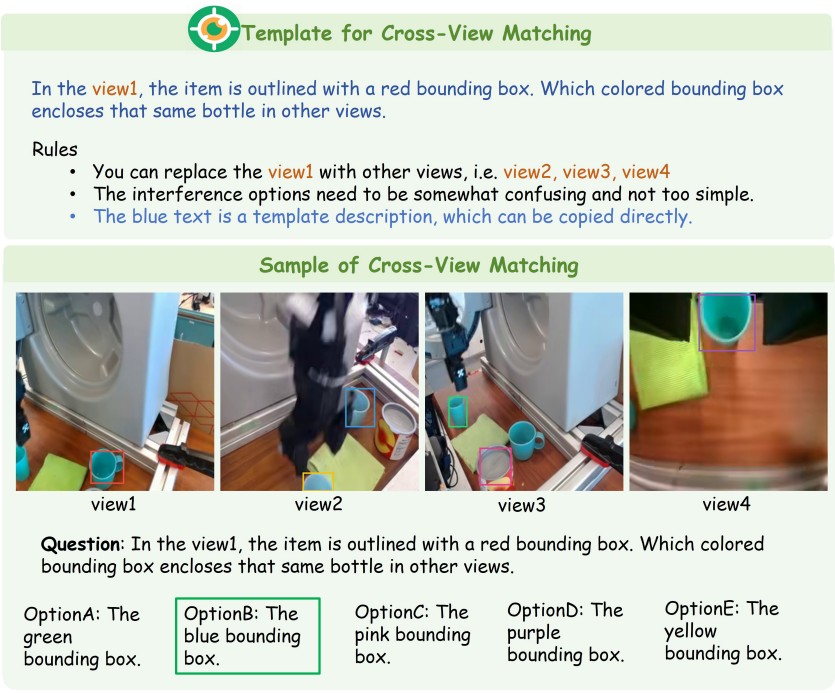

Figure 15: Example of *Cross-View Object Matching* constructed from the BridgeV2 dataset. The target is highlighted in view1, and the model must select the corresponding bounding box in the other synchronized views.

To ensure non-triviality, distractor options are manually verified so that objects with similar 2D appearances may differ in their actual 3D distances. Accurate solutions therefore demand reasoning that goes beyond single-view perception.

Figures 16 and 17 illustrate both representative instances and the annotation templates employed for constructing the *Distance Judgement* subtask in AgiWorld and BridgeV2.

### F.3    VIEWPOINT IDENTIFICATION

This subtask belongs to the **spatial** category and evaluates a model's ability to recognize and reason across different viewpoints. It is constructed exclusively from the AgiWorld dataset, where the reference image is always taken from the head camera. The question asks the model to determine which candidate image corresponds to the correct left- or right-gripper view at the same time step, given the head camera observation. Solving the task requires a form of perspective transformation, testing whether the model can imagine how the scene would appear from another viewpoint—a core component of spatial intelligence.

To construct distractor options, we adopt a multi-stage design. For each ground truth gripper image, we first include the image from the opposite gripper at the same time step. We then add distractors from the same episode but different time steps, ensuring a non-trivial temporal gap in the gripper poses so that the distractor cannot be rejected trivially. Additional distractors are sampled from other episodes within the same or closely related tasks, providing visually plausible but incorrect gripper views. This strategy prevents the model from exploiting majority-vote heuristics across options. All instances are manually verified to guarantee that a human annotator can reliably identify the correct correspondence based on spatial details.

Figure 18 illustrates both the template and an example instance of this subtask.

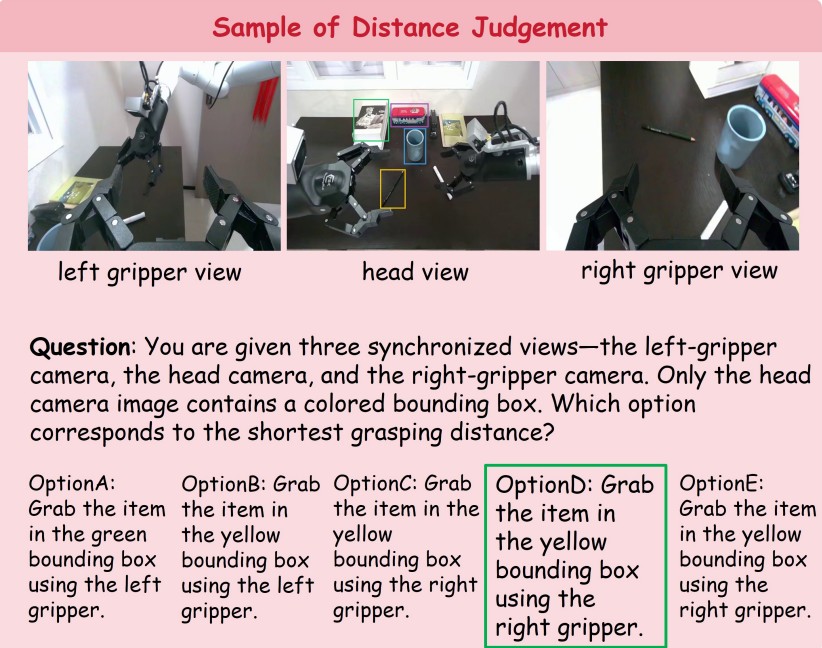

**Template for Distance Judgement**

You are given three synchronized views—the left-gripper camera, the head camera, and the right-gripper camera. Only the head camera image contains colored bounding boxes. Which option corresponds to the shortest grasping distance?

Rules
- The orange box is a good interference option because there is depth ambiguity in the head camera view.
- You can replace the shortest with longest.
- The blue text is a template description, which can be copied directly.

**Sample of Distance Judgement**

left gripper view          head view          right gripper view

**Question**: You are given three synchronized views—the left-gripper camera, the head camera, and the right-gripper camera. Only the head camera image contains a colored bounding box. Which option corresponds to the shortest grasping distance?

OptionA: Grab the item in the green bounding box using the left gripper.

OptionB: Grab the item in the yellow bounding box using the left gripper.

OptionC: Grab the item in the yellow bounding box using the right gripper.

OptionD: Grab the item in the yellow bounding box using the right gripper.

OptionE: Grab the item in the yellow bounding box using the right gripper.

Figure 16: Example of *Distance Comparison* constructed from the AgiWorld dataset. The head camera image contains candidate bounding boxes, and the model must select the one corresponding to the shortest grasping distance.

## F.4 3D Spatial Consistency

This subtask is part of the **spatial** category and evaluates a model's ability to reason about object locations within a structured 3D coordinate system. The key challenge is to assess whether the model can treat the scene as a three-dimensional space rather than a flat image, and correctly place the highlighted objects into the standardized coordinate grid such that their relative positions remain coherent across views.

We adopt a right-handed orthogonal coordinate system anchored to a designated reference view (the head camera in AgiWorld, or any of the four views in BridgeV2). In the reference image, several target objects are highlighted with colored bounding boxes. The question then asks the model: *"Which of the following sets of coordinate triplets best describes the positions of the highlighted objects?"* Coordinates are normalized into a $5 \times 5 \times 5$ cubic grid, with integer values from 1 to 5 along each axis. This abstraction allows spatial relations to be expressed consistently without requiring precise metric depth.

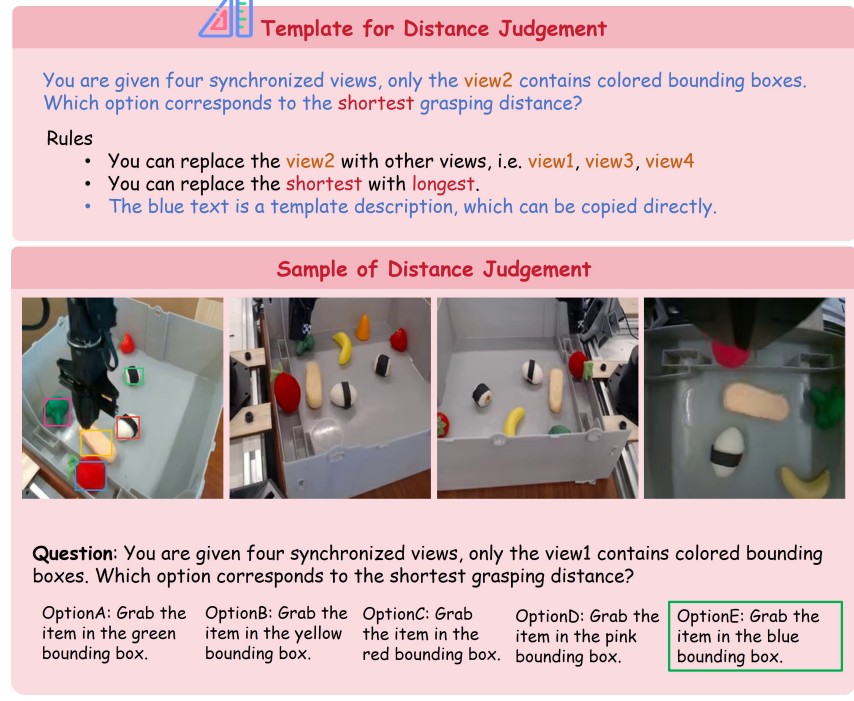

Figure 17: Example of *Distance Comparison* constructed from the BridgeV2 dataset. One view contains candidate bounding boxes, and the model must identify the option that corresponds to the shortest grasping distance when integrating evidence across all four views.

To construct the tasks, we leverage the interactive cube visualization tool described in Appendix E.4. This tool enables annotators to map each object to a unit cube in the grid, adjust placements, and generate candidate coordinate sets. Distractor options are created by perturbing object coordinates to introduce plausible but incorrect spatial configurations. Accurate solutions therefore require integrating multi-view cues rather than relying on a single perspective.

Figures 19 and 20 show representative templates and examples constructed from the AgiWorld and BridgeV2 datasets, respectively.

### F.5 ACTION PLANNING

This subtask belongs to the **robotic** category and evaluates whether a model can correctly identify the valid high-level action sequence from multiple candidates in order to accomplish a manipulation goal. Each instance provides synchronized multi-view observations together with a task description in natural language. The problem is defined with respect to a designated reference view, within which we establish the standardized right-handed coordinate system described in Appendix E.3. Accordingly, all candidate action sequences are expressed as sequences of normalized directional terms (i.e., spatial adverbs such as *leftward*, *forward*, *downward*), which follow directly from the axis conventions defined in Appendix E.3. The model must then integrate information across views and select the sequence most likely to achieve the goal.

To ensure non-triviality, distractor options are carefully constructed. Only one option corresponds to a valid sequence that completes the task while minimizing collisions, whereas the distractors follow plausible but incorrect paths. In addition, we enumerate and sort the directional terms within each option, ensuring that no two candidates share the same ordered sequence of actions. This design prevents ambiguity and forces the model to reason jointly about spatial relations and manipulation feasibility.

Figures 21 and 22 illustrate representative templates and examples from the AgiWorld and BridgeV2 datasets, respectively.

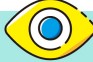 **Template for Viewpoint Identification**

Given the image captured by the head camera, which of the following images shows the right-gripper camera's view at that exact moment?

Rules
- You can replace the right-gripper with left-gripper.
- The blue text is a template description, which can be copied directly.

**Sample for Viewpoint Identification**

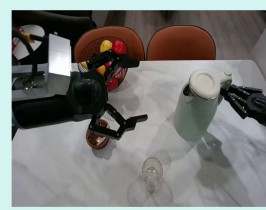

**Question**: Given the image captured by the head camera, which of the following images shows the left gripper camera's view at that exact moment?

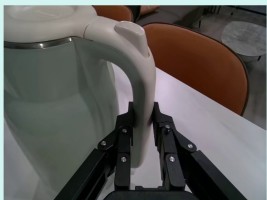

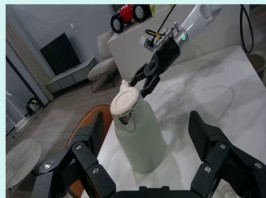

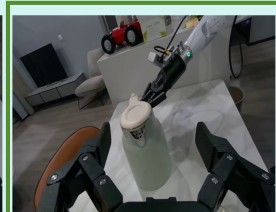

OptionA: Option A picture.

OptionB: Option B picture.

OptionC: Option C picture.

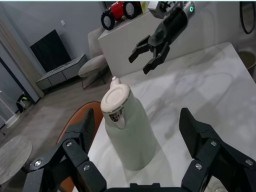

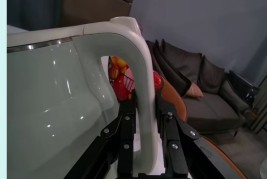

OptionD: Option D picture.

OptionE: Option E picture.

Figure 18: Example of *Perspective Identification* constructed from the AgiWorld dataset. The head camera view is used as the reference, and the model must infer the correct corresponding perspective among the candidate gripper views.

## F.6 STEP EXECUTION

This subtask belongs to the **robotic** category and focuses on low-level action execution in manipulation tasks. Each instance provides synchronized multi-view observations together with a natural language description of the goal. Unlike the *Action Planning* task, which evaluates multi-step trajectories, *Step Execution* concentrates on primitive actions such as picking or placing, which can be described as short sequences of directional terms (e.g., *up*, *left*, *down*). The coordinate system is defined with respect to a designated reference view, following the conventions introduced in Ap-

pendix E.3. All candidate options are then expressed in these normalized directional terms, and the model must select the sequence that correctly achieves the task.

Distractor options are constructed to appear plausible but correspond to incorrect motions that would fail the manipulation. To eliminate redundancy, we further enumerate and sort the directional terms within each option, ensuring that no two candidates reduce to the same ordered sequence. This design requires the model to interpret spatial cues accurately across multiple views and to ground its decision in the standardized coordinate system. For the AgiWorld dataset, the template is based on synchronized left-gripper, head, and right-gripper views, while in BridgeV2 any of the four available views may serve as the reference.

Figures 23 and 24 show representative templates and examples from the AgiWorld and BridgeV2 datasets, respectively.

### F.7    TRAJECTORY SELECTION

This subtask belongs to the **robotic** category and evaluates a model's ability to reason about complete motion trajectories in multi-view settings. Each instance provides synchronized observations, where candidate trajectories are overlaid in different colors on one or more reference views. The model is asked to determine which trajectory is most likely to accomplish the described manipulation.

A key challenge is that trajectories drawn in a single view may be ambiguous due to occlusions, perspective distortion, or motion along the camera's optical axis. By providing multiple synchronized viewpoints, the task requires the model to integrate cross-view evidence to correctly identify the feasible trajectory.

All distractor trajectories are *manually curated* to be distinct from the ground truth yet visually plausible, so that they may appear confusing at first glance but remain distinguishable through careful multi-view reasoning. We ensure that exactly one candidate is feasible across views and can complete the task without collisions; every instance is human-validated to confirm that the correct choice is uniquely identifiable.

For the AgiWorld dataset, each problem is presented with synchronized left-gripper, head, and right-gripper views. For BridgeV2, all four camera perspectives are available, and candidate trajectories are described relative to these views. Figures 25 and 26 provide representative templates and examples from both datasets.

### F.8    AFFORDANCE RECOGNITION

This subtask belongs to the **robotic** category and evaluates a model's ability to recognize feasible grasp candidates in multi-view scenes. In real manipulation, a single viewpoint may be insufficient for identifying good grasp locations due to occlusions by objects or grippers, or because certain camera angles (e.g., top-down) obscure critical contact geometry. By incorporating synchronized multi-view observations, especially from gripper-mounted cameras, this task provides complementary perspectives that make the final grasp point more reliably observable.

Each instance presents five candidate grasps illustrated with color-coded lines (red, yellow, green, blue, and pink). Each color appears exactly once across the available views, and the two endpoints of a line specify the intended positions of the gripper fingers. The model is asked: *"Which color-coded line represents the grasp candidate most likely to succeed?"*

All distractors are carefully designed: while they may appear physically plausible at first glance, they are infeasible in practice due to orientation, collision risk, or instability. This ensures that success requires genuine spatial reasoning and affordance understanding rather than superficial cues. For the AgiWorld dataset, three views (left-gripper, head, right-gripper) are used, whereas in BridgeV2 the template extends naturally to four synchronized views. Figures 27 and 28 provide representative templates and examples from both datasets.

Table 6: Overview of the eight subtasks in our benchmark. Spatial tasks focus on multi-view scene understanding, while robotic tasks extend this foundation to manipulation planning and execution.

| Category | Subtask | Core Ability Assessed |
|---|---|---|
| Spatial | Cross-View Object Matching | Identify the same object across different viewpoints despite distractors. |
| | Distance Judgement | Compare relative distances to a specified gripper using multi-view cues. |
| | Viewpoint Identification | Infer the correct camera perspective given a head-view reference. |
| | 3D Spatial Consistency | Place highlighted objects into a structured 3D coordinate system with coherent relative positions. |
| Robotic | Action Planning | Select the valid high-level action sequence in normalized directional terms to accomplish a task. |
| | Step Execution | Choose the correct primitive low-level action sequence (e.g., pick/place) grounded in the coordinate system. |
| | Trajectory Selection | Distinguish feasible from infeasible motion trajectories by integrating evidence across views. |
| | Affordance Recognition | Identify the grasp candidate most likely to succeed among visually plausible alternatives. |

## F.9 ANSWER BALANCING AND RANDOMIZATION

After generating QA instances and completing manual verification, we apply an additional balancing step to ensure that answer distributions are statistically uniform. Specifically, correct answers are randomized across different option indices and color assignments, preventing systematic biases that could allow models to exploit position- or color-based heuristics. This balancing guarantees that success on the benchmark requires genuine spatial reasoning and affordance understanding rather than relying on superficial answer patterns.

## F.10 SUMMARY OF BENCHMARK CONSTRUCTION

Taken together, the eight subtasks provide a comprehensive evaluation of spatial and robotic reasoning in multi-view environments.

The four **robotic** subtasks (Action Planning, Step Execution, Trajectory Selection, and Affordance Recognition) extend this foundation to manipulation scenarios. They examine whether models can ground spatial understanding into action decisions, ranging from high-level planning to low-level execution, and from trajectory-level reasoning to grasp affordance prediction. Together, they highlight the importance of combining multi-view perception with physical feasibility in order to succeed in robotic tasks.

An overview of all subtasks, their categories, and the specific reasoning abilities they target is provided in Table 6.

# G ABLATION STUDY ON MULTI-VIEW NECESSITY

In this section, we address the fundamental question of whether multi-view inputs are truly necessary for the proposed robotic reasoning tasks, or if a single-view input would suffice. To investigate this, we conducted a systematic ablation study comparing the performance of representative models under a **Single-View** baseline versus the standard **Multi-View** setting.

## G.1 TASK SELECTION AND EXPERIMENTAL SETUP

Our benchmark consists of eight subtasks. However, not all tasks are suitable for single-view evaluation due to their inherent reliance on cross-view information. We applied the following selection criteria:

- **Excluded Tasks (Inherently Multi-View):** For subtasks including *Cross-View Object Matching*, *Viewpoint Identification*, *Trajectory Selection*, and *Affordance Recognition*, the question semantics inherently depend on multiple synchronized views. Many answer candidates exist only in specific views, so removing views would yield ill-posed questions (e.g., some candidates become invisible). Therefore, these tasks were excluded from the ablation.

- **Selected Tasks (Degradable to Single-View):** We focused our ablation on the remaining four subtasks: *Distance Judgement*, *3D Spatial Consistency*, *Action Planning*, and *Step Execution*. These tasks are typically formulated relative to a reference coordinate system or a primary scene description. While multi-view information provides critical depth cues and occlusion handling, these questions remain logically valid even when restricted to a single input image. This allows us to rigorously measure the "performance drop" caused by the loss of multi-view context.

For the **Single-View** setting, we retained only the most informative third-person perspective to ensure a strong baseline:

- For the **AgiWorld** dataset, we used the *head camera* view.
- For the **BridgeV2** dataset, we used *view1* (a fixed third-person camera).

## G.2 RESULTS AND ANALYSIS

Table 7 presents the comparative results. We report the full multi-view accuracy and the performance gap ($\Delta$) relative to the single-view baseline.

Table 7: Comparison of Single-View vs. Multi-View performance on selected subtasks. The values represent Multi-View accuracy, and values in parentheses indicate the change ($\Delta$) compared to the Single-View baseline. Positive $\Delta$ indicates that multi-view inputs improve performance. **Bold** indicates the best performance in each category.

| Model | Avg. | Distance Judge. | 3D Spatial Cons. | Action Plan. | Step Exec. |
|---|---|---|---|---|---|
| *Proprietary Reasoning Models* | | | | | |
| GPT-5 | **64.65 (+6.69)** | **36.32 (+18.90)** | **78.43 (+3.92)** | **76.96 (+2.45)** | **66.24 (+2.14)** |
| GPT-5-chat | 28.11 (+3.05) | 29.35 (+13.44) | 10.29 (-5.39) | 35.29 (+1.47) | 36.32 (+3.85) |
| *Proprietary Models* | | | | | |
| GPT-4.1 | 24.20 (+3.59) | **32.84 (+10.44)** | 4.90 (+1.47) | 30.39 (-0.49) | 28.21 (+3.41) |
| GPT-4o | **26.81 (+0.88)** | 31.84 (+5.47) | **5.39 (+0.98)** | **33.82 (-0.49)** | **35.04 (-1.28)** |
| *Open-Source Models* | | | | | |
| Qwen2.5-vl-72b | **23.01 (+0.87)** | **30.85 (+3.98)** | 3.92 (+0.98) | **30.88 (-2.45)** | **26.07 (+1.28)** |
| Qwen2.5-vl-32b | 20.28 (-0.07) | 24.38 (+1.49) | **8.33 (+2.45)** | 25.49 (-1.98) | 22.65 (-2.99) |
| Qwen2.5-vl-7b | 17.91 (+1.55) | 20.40 (0.00) | 4.90 (+3.92) | 21.08 (+1.47) | 24.36 (+1.71) |

Our analysis yields three key findings regarding the necessity of multi-view perception:

1. **Multi-view is critical for resolving spatial ambiguity.** The most significant impact is observed in the *Distance Judgement* task. Powerful reasoning models like GPT-5 and GPT-4.1 achieve substantial gains (+18.90% and +10.44%, respectively) when provided with multi-view inputs. This confirms that single-view observations suffer from inherent depth ambiguity and occlusion—common issues in robotic manipulation—which are effectively mitigated by integrating complementary viewpoints.

2. **Stronger reasoning capabilities unlock multi-view potential.** We observe a positive correlation between model capability and the benefit derived from multi-view information. State-of-the-art models consistently improve with additional views, whereas smaller models (e.g., Qwen2.5-vl-32b, Qwen2.5-vl-7b) show negligible or even negative changes. This suggests that effectively fusing discordant visual information requires strong spatial reasoning capabilities; without this, smaller models may be distracted by the increased visual context rather than aided by it.

3. **Performance gap remains significant.** Even with the advantage of multi-view inputs, the performance on robotic planning tasks generally lags behind human proficiency. This highlights that while multi-view input is a necessary condition for robust perception, it is not sufficient on its own. Future models must develop stronger embodied reasoning capabilities to fully leverage the rich 3D information provided by MV-RoboBench.

# H    IMPACT OF IMAGE ORIENTATION ON SPATIAL REASONING

To evaluate the generalization capabilities of VLMs regarding image orientation, we conducted an additional stress test. In this experiment, we vertically flipped **all camera views except for the gripper-mounted ones** (e.g., the head camera in AgiWorld and all static third-person views in BridgeV2) upside down. This setup disrupts the canonical spatial structure (e.g., visual "up" no longer aligns with gravity) without altering the intrinsic object relationships.

**Crucially, this manipulation does not affect the correctness of the ground truth answers.** As defined in Appendix E.3, our coordinate systems are established based on the physical gravity vector ($g$) and the camera's optical axis ($c$), rather than the 2D image pixel axes. Since the physical scene configuration and sensor properties remain unchanged, the logical spatial relationships (e.g., "left", "up", "forward") remain valid. Therefore, any performance drop can be attributed solely to the models' inability to generalize to non-canonical visual orientations.

Table 8 reports the detailed performance under this upside-down setting across all subtasks.

Table 8: Detailed performance under Upside-Down image orientation. "**Avg**" represents the average accuracy in the upside-down setting. "Δ" denotes the performance drop compared to the original setting. The remaining columns show the specific accuracy for each subtask under the upside-down condition.

| Model | Avg | Δ | Spatial Tasks | | | | Robotic Tasks | | | |
| --- | --- | --- | --- | --- | --- | --- | --- | --- | --- | --- |
| | | | Cross | Dist. | View. | 3D Cons. | Plan | Step | Traj. | Afford. |
| *Proprietary Reasoning Models* | | | | | | | | | | |
| GPT-5 | **37.68** | **-18.73** | 33.50 | 30.35 | 25.00 | 26.47 | 47.55 | 40.60 | 53.50 | 44.50 |
| GPT-5-mini | 30.79 | -7.49 | 31.50 | 23.88 | 21.88 | 17.16 | 46.57 | 38.89 | 42.50 | 23.92 |
| GPT-5-nano | 22.42 | -10.33 | 17.00 | 18.41 | 22.66 | 11.76 | 28.43 | 25.21 | 30.50 | 25.36 |
| GPT-5-chat | 28.50 | -3.13 | 31.00 | 35.32 | 27.34 | 6.37 | 33.33 | 33.76 | 35.50 | 25.36 |
| *Proprietary Models* | | | | | | | | | | |
| GPT-4.1 | 26.72 | -4.18 | 26.50 | 33.33 | 33.59 | 4.90 | 27.94 | 29.49 | 37.00 | 21.05 |
| GPT-4o | 25.10 | -2.49 | 25.50 | 31.84 | 23.44 | 4.90 | 30.39 | 31.20 | 33.00 | 20.57 |
| GPT-4.1-mini | 23.94 | -0.04 | 22.50 | 26.37 | 27.73 | 6.37 | 25.98 | 25.21 | 32.50 | 24.88 |
| GPT-4.1-nano | 20.08 | -0.77 | 17.50 | 16.42 | 18.36 | 12.25 | 25.98 | 25.21 | 20.00 | 24.88 |
| *Open-Source Models* | | | | | | | | | | |
| Qwen2.5-vl-72b | 23.33 | -0.96 | 23.50 | 23.88 | 22.27 | 3.92 | 28.92 | 27.35 | 29.50 | 27.27 |
| Qwen2.5-vl-32b | 22.41 | -0.07 | 24.50 | 28.86 | 23.05 | 3.92 | 25.00 | 23.93 | 28.50 | 21.53 |
| Qwen2.5-vl-7b | 19.71 | -1.13 | 22.00 | 20.90 | 20.31 | 5.39 | 23.04 | 26.50 | 19.00 | 20.57 |

We observe distinct behaviors across model families:

1. **Strong models exhibit strong orientation bias.** The most capable model, GPT-5, suffers the largest performance drop (Δ = -18.73%). Comparing this to its baseline performance in Table 2, we see a dramatic decline in *3D Spatial Consistency* (from 82.35% to 26.47%) and *Distance Judgement* (from 55.22% to 30.35%). This indicates that its superior spatial reasoning capabilities rely heavily on the canonical "upright" orientation learned during pre-training. When the visual reference frame is inverted, the model struggles to maintain coherent 3D spatial relations.

2. **Weaker models show a floor effect.** In contrast, smaller models (e.g., GPT-4.1-mini, Qwen2.5-vl-7b) show negligible performance changes (Δ ≈ 0). As shown in Table 2, these models already perform poorly on spatial tasks in the standard setting (e.g., ~7-9% on *3D Spatial Consistency*). This confirms that their original performance was likely dominated by pattern matching or random guessing rather than genuine spatial understanding, making them insensitive to orientation flips.

3. **Implications for deployment.** While this result highlights a limitation in zero-shot generalization, we note that in real-world robotic deployment, camera mounting poses are strictly controlled, and input images are typically inspected or rectified to ensure a canonical orientation. Therefore, while VLMs lack rotation invariance, this sensitivity is a manageable characteristic in engineered systems.

# I  EVALUATION OF MODEL CAPABILITY TO REJECT INCORRECT OPTIONS

The ability to determine when no correct answer exists is critical for safe robotic deployment. As pointed out by the reviewers, evaluating this "rejection" capability is a vital aspect of spatial reasoning.

In our original benchmark design, we partially incorporated this dimension. Specifically, in the *Cross-View Object Matching* task, approximately 25% of the questions explicitly include a ground-truth option of "None of the bounding boxes is correct." This was designed to prevent models from relying solely on elimination.

However, to provide a more comprehensive and extreme evaluation of this capability across the entire benchmark, we conducted a controlled stress test on 7 out of the 8 subtasks.

## I.1  EXPERIMENTAL SETUP

We modified the dataset by replacing the original ground-truth option with "None of the above" for every question in the selected subtasks. The original correct answer was removed, making "None of the above" the only valid choice. To avoid positional bias, we ensured this option appeared as option E.

**Exclusion of Distance Judgement:** We excluded the *Distance Judgement* task from this specific ablation. In this task, questions ask for the "shortest" or "longest" distance among candidates. Removing the absolute shortest candidate would logically make the *second* shortest candidate the new correct answer, creating ambiguity rather than a clear "None of the above" scenario. All other 7 subtasks allow for a binary valid/invalid distinction, making them suitable for this test.

## I.2  RESULTS AND ANALYSIS

Table 9 presents the results. We report the accuracy when the correct answer is "None of the above" (Reject. Avg.) and the performance drop compared to the standard setting.

Table 9: Model performance on the "None of the above" rejection test. "**Reject.**" denotes the accuracy on the modified dataset where the correct answer is "None of the above". "**Drop**" indicates the performance decline relative to the original benchmark accuracy. A significant drop implies high model over-compliance.

| Model | Reject. Avg. | Drop | Spatial Tasks | | | Robotic Tasks | | | |
| --- | --- | --- | --- | --- | --- | --- | --- | --- | --- |
| | | | Cross-View Match | Viewpoint ID | 3D Spatial Consist. | Action Plan. | Step Exec. | Trajectory Sel. | Affordance Rec. |
| *Proprietary Reasoning Models* | | | | | | | | | |
| GPT-5 | **13.43** | -43.29 | 29.00 | 6.64 | 6.37 | 23.04 | 20.09 | 6.00 | 2.87 |
| GPT-5-mini | 9.57 | -34.71 | 26.00 | 0.39 | 9.80 | 8.33 | 20.09 | 0.50 | 1.91 |
| GPT-5-chat | 1.05 | -28.84 | 1.50 | 1.95 | 1.47 | 0.98 | 0.43 | 1.00 | 0.00 |
| GPT-5-nano | 8.61 | -24.02 | 21.00 | 5.08 | 16.18 | 2.94 | 7.26 | 3.50 | 4.31 |
| *Proprietary Models* | | | | | | | | | |
| GPT-4.1 | 0.60 | -27.35 | 1.50 | 0.78 | 0.98 | 0.49 | 0.43 | 0.00 | 0.00 |
| GPT-4.1-mini | 3.74 | -19.09 | 3.00 | 1.95 | 1.47 | 3.92 | 15.81 | 0.00 | 0.00 |
| GPT-4.1-nano | 0.78 | -18.21 | 0.00 | 0.00 | 0.00 | 0.00 | 2.56 | 0.50 | 2.39 |
| GPT-4o | 0.92 | -23.51 | 1.50 | 3.12 | 0.49 | 0.00 | 0.85 | 0.50 | 0.00 |
| *Open-Source Models* | | | | | | | | | |
| Qwen2.5-vl-72b | 3.39 | -19.81 | 9.00 | 0.78 | 8.82 | 1.96 | 1.71 | 1.00 | 0.48 |
| Qwen2.5-vl-7b | 0.49 | -20.31 | 2.00 | 0.00 | 0.98 | 0.00 | 0.00 | 0.00 | 0.48 |
| Qwen2.5-vl-32b | 0.28 | -21.70 | 1.00 | 0.00 | 0.49 | 0.00 | 0.00 | 0.00 | 0.48 |

The results reveal a severe "Over-Compliance" phenomenon across all models:

1. **Universal collapse in rejection capability.** Even the strongest model, GPT-5, drops from 56% accuracy to  13% when forced to reject incorrect options. Most other models collapse to near 0%, indicating they almost always hallucinate a relationship or force a selection from the visual candidates rather than acknowledging the absence of a correct answer.

2. **Selection bias dominates validity judgement.** The experiment demonstrates that current models exhibit a strong tendency to select a "visually approximate" option rather than re-

jecting all options. Choosing "None of the above" imposes a higher cognitive requirement: the model must distinguish between "relative similarity" and "absolute correctness," and possess an internal standard of rationality to invalidate all candidates. The results confirm that even advanced reasoning models currently struggle with this rigorous verification, preferring to conform to the prompt by picking a plausible-looking but incorrect answer.

3. **Implication.** This experiment highlights a critical safety gap in current VLM technology. While models can reason about what is present, they struggle significantly to verify correctness by rejecting invalid options. MV-RoboBench effectively exposes this limitation, serving as a testbed for future work on uncertainty estimation and refusal.

## J  ERROR ANALYSIS OF MODEL FAILURES

To better understand how current models fail on MV-RoboBench, we conduct a qualitative error analysis on three representative models: a strong proprietary model (GPT-5), a strong open model (Qwen2.5-VL-72B-Instruct), and a weaker open model (Qwen2.5-VL-7B-Instruct). For each model, we sample 10–20 erroneous cases from several core subtask families, and manually inspect their predictions and explanations. Below we summarize the main recurring failure modes and highlight representative examples.

**Single-view bias and weak multi-view fusion.**   Across all three models, a prominent failure mode is an over-reliance on a single camera view while largely ignoring contradictory evidence from other views. GPT-5, for instance, often selects the option that looks most natural in the head view but becomes inconsistent once the gripper views are taken into account. Qwen2.5-VL-7B shows an even stronger single-view bias in cross-view matching and distance judgment tasks, where it tends to align 2D screen positions across cameras (e.g., "bottom-right" to "bottom-right") instead of reasoning in a shared 3D frame. Qwen2.5-VL-72B behaves similarly in viewpoint identification and cross-view tasks, effectively treating them as single-image retrieval problems rather than enforcing multi-view geometric consistency.

**Depth, occlusion, and 3D geometry confusion.**   A second common failure mode involves incorrect reasoning about depth, occlusion, and 3D layout. In GPT-5, distance judgment and 3D spatial consistency errors show that the model frequently substitutes 2D heuristics (apparent size, 2D position) for true 3D reasoning: it chooses objects that appear more salient or closer in a single projection, even when other views clearly indicate that another object is nearer in 3D. Qwen2.5-VL-7B exhibits even stronger depth confusion, sometimes inverting near–far or front–back relations, and occasionally treating objects on the tabletop and objects below the table (e.g., cabinet handles) as comparable candidates. Qwen2.5-VL-72B shows similar issues in fine-grained 3D spatial consistency, where it misassigns grid coordinates along the depth axis or swaps two neighboring objects that are close in distance. These patterns suggest that current models lack robust internal 3D scene representations and often rely on weak perspective cues.

**Frame-of-reference errors and instance-level correspondence.**   Even with an explicitly defined camera-centric coordinate system (Appendix E.3), models often mishandle frame-of-reference details and instance-level matching. For GPT-5, viewpoint identification and cross-view matching failures indicate that the model sometimes implicitly assumes that the same set of objects must be visible in all views, and treats changes in visibility due to field-of-view or occlusion as evidence that the views are asynchronous or unrelated. For Qwen2.5-VL-7B, we frequently observe instance-level confusion in scenes with multiple objects of the same category (e.g., several yellow peppers): the model correctly recognizes the category but fails to track which specific instance is highlighted across views, effectively performing category-level rather than instance-level correspondence. Qwen2.5-VL-72B exhibits similar behavior in multi-object setups, where it matches "any object of the correct type" instead of the exact instance indicated by the bounding box. These errors highlight the difficulty of precise, instance-level multi-view alignment even for large models.

**Affordance and physical-feasibility misunderstandings.**   In affordance recognition and trajectory selection tasks, models must reason about which grasps or motion paths are more likely to succeed physically. GPT-5 often prefers grasp lines or trajectories that look visually neat (e.g., centered on the visible surface) but would be unstable or collision-prone for a parallel gripper in 3D; it

rarely reasons explicitly about approach direction, object thickness, or nearby obstacles. Qwen2.5-VL-7B sometimes mislocalizes the grasp line to the wrong object altogether and then optimizes within this incorrect frame, leading to explanations that sound plausible but are grounded in the wrong spatial reference. Qwen2.5-VL-72B exhibits related issues: it often treats "passing through the middle" as a universal heuristic for good paths, even when the task requires side grasps on thin objects or collision-aware trajectories that avoid table edges and surrounding clutter. Overall, these failures indicate that current VLMs still lack robust modeling of robotic affordances and physical constraints, especially when such reasoning must be carried out jointly with multi-view geometric alignment.

**Summary.**  Importantly, these failure modes appear not only in weaker models but also in GPT-5, which still exhibits systematic errors in multi-view fusion, depth reasoning, frame-of-reference handling, and affordance understanding. This suggests that MV-RoboBench does not merely lower raw accuracy; it exposes non-trivial limitations in current vision–language models' spatial reasoning, and can serve as a diagnostic tool for guiding future architectural and training improvements. While human-centric research like EgoBrain (Lin et al., 2025a) focuses on decoding intentions, and methods exploring diffusion guidance and preference alignment (Chen et al., 2025b;a) successfully enhance the quality and coherence of generative models, our findings highlight that current VLMs still lack the physical spatial reasoning required for successful robotic execution.

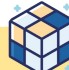
### Template for 3D Spatial Consistency

You are given three synchronized views—the left-gripper camera, the head camera, and the right-gripper camera. Only the head camera image contains colored bounding boxes. Then assuming the workspace is divided into a 5 × 5 × 5 cubic grid, with coordinates (x, y, z) running from 1 to 5. Which of the following sets of coordinate triplets best describes the locations of the objects circled by the bounding boxes in the image above? The following options are based on the head camera view. + <Description of coordinate system definition>

Rules
- The blue text is a template description, which can be copied directly.

### Sample of 3D Spatial Consistency

| left gripper view | head view | right gripper view |
|---|---|---|

**Question**: As shown above

OptionA: (Red box, 2, 4, 4), (Yellow box, 5, 3, 4), (Green box, 1, 4, 1), (Blue box, 4, 3, 4), (Pink box, 4, 3, 2)

OptionB: (Red box, 2, 4, 4), (Yellow box, 4, 4, 4), (Green box, 1, 4, 1), (Blue box, 4, 3, 4), (Pink box, 4, 3, 2)

OptionC: (Red box, 1, 4, 4), (Yellow box, 5, 3, 4), (Green box, 2, 4, 1), (Blue box, 4, 3, 4), (Pink box, 4, 3, 2)

OptionD: (Red box, 2, 4, 4), (Yellow box, 4, 4, 4), (Green box, 3, 4, 1), (Blue box, 1, 2, 3), (Pink box, 4, 3, 2)

OptionE: (Red box, 1, 4, 4), (Yellow box, 5, 3, 4), (Green box, 2, 4, 1), (Blue box, 4, 3, 2), (Pink box, 4, 3, 4)

Figure 19: Example of *Spatial Cube Reasoning* from the AgiWorld dataset. Objects are localized in a 5 × 5 × 5 cubic grid, and the model must select the correct coordinate triplets from the given options.

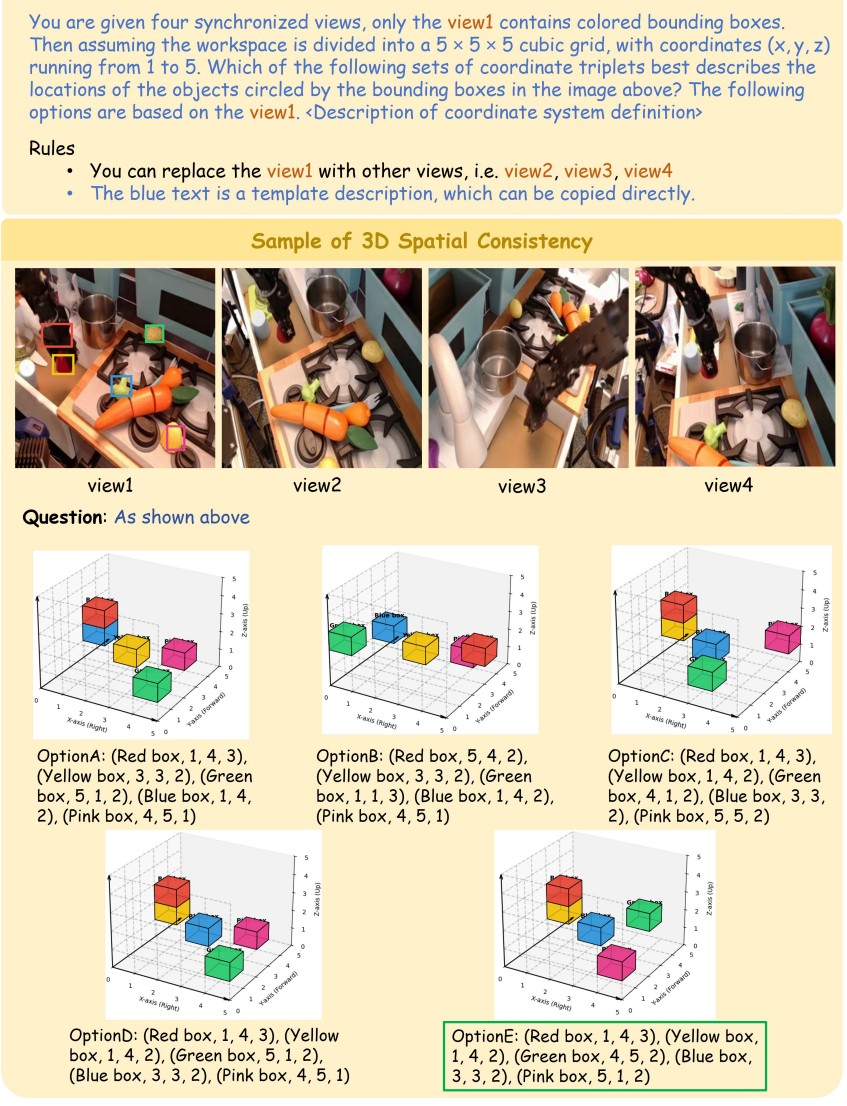

Figure 20: Example of *Spatial Cube Reasoning* from the BridgeV2 dataset. One reference view (here, `view1`) contains bounding boxes, and the model must infer the correct 3D coordinates of the objects across the synchronized views.

## 📋⏰ Template for Action Planning

**\<Task\>** Which of the following operations is most likely to complete the task with the least collision? The following options are based on the head camera view. \<Description of coordinate system definition\>

Rules
- You can replace the head with left-gripper or right-gripper.
- The blue text is a template description, which can be copied directly.

## Sample of Action Planning

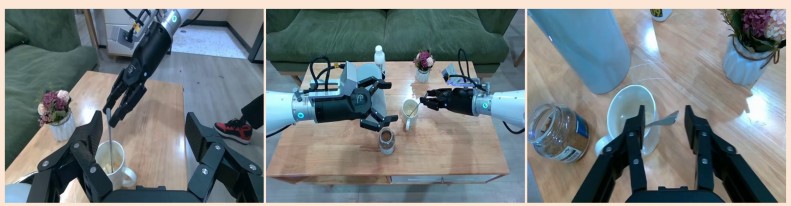

left gripper view       head view       right gripper view

**Question**: If I want to pour water from the kettle on the table into the water cup. \<Template which shown as above\>

OptionA: Move the left gripper leftward, then forward, then downward to grasp the handle of the kettle. Then lift the kettle, move it upward, and then rightward to pour water into the cup.

OptionB: Move the left gripper rightward, then backward, then upward to grasp the handle of the kettle. Then lower the kettle, move it downward, and then leftward to pour water into the cup.

OptionC: Move the left gripper downward, then rightward, then forward to grasp the handle of the kettle. Then lift the kettle, move it forward, and then leftward to pour water into the cup.

OptionD: Move the left gripper forward, then leftward, then upward to grasp the handle of the kettle. Then lift the kettle, move it rightward, and then downward to pour water into the cup.

OptionE: Move the left gripper backward, then upward, then leftward to grasp the handle of the kettle. Then lift the kettle, move it downward, and then forward to pour water into the cup.

Figure 21: Example of *Planning* from the AgiWorld dataset. The model must select the correct sequence of normalized directional actions to accomplish the described task with minimal collisions.

📋🕐 **Template for Action Planning**

**\<Task\>** Which of the following operations is most likely to complete the task with the least collision? The following options are based on the view2. \<Description of coordinate system definition\>

Rules
- You can replace the view2 with other views, i.e. view1, view3, view4
- The blue text is a template description, which can be copied directly.

**Sample of Action Planning**

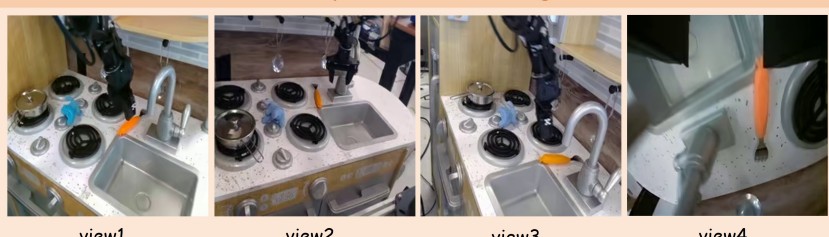

view1          view2          view3          view4

**Question**: I want to put the carrot from the stove into the sink. Which of the following operations is most likely to complete the task with the least collision? The following options are based on the view2.\<Template which shown as above\>

OptionA: Move the gripper to the left, then upward, then backward to grab, then left, then backward, then right, and finally upward to put the carrot into the sink to complete the operation.

OptionB: Move the gripper to the left, then forward, then downward to grab, then upward, then backward, then right, and finally downward to put the carrot into the sink to complete the operation.

OptionC: Move the gripper to the left, then forward, then downward to grab, then forward, then backward, then right, and finally upward to put the carrot into the sink to complete the operation.

OptionD: Move the gripper to the left, then forward, then downward to grab, then left, then backward, then right, and finally upward to put the carrot into the sink to complete the operation.

OptionE: Move the gripper to the left, then backward, then downward to grab, then left, then backward, then right, and finally upward to put the carrot into the sink to complete the operation.

Figure 22: Example of *Planning* from the BridgeV2 dataset. One reference view (here, `view2`) is used to describe the options, and the model must infer the correct high-level sequence to achieve the task while avoiding collisions.

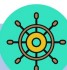 **Template for Step Execution**

**<Task>,** which of the following actions is most likely to accomplish this task? The following options are based on the head camera view. <Description of coordinate system definition>

Rules
- You can replace the head with left-gripper or right gripper.
- The blue text is a template description, which can be copied directly.

**Sample for Step Execution**

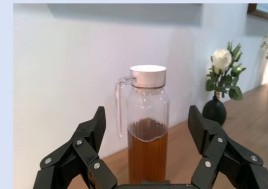 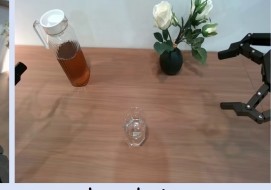 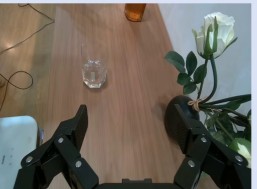

left gripper view          head view          right gripper view

**Question**: Suppose I want to use the right gripper to grab the spoon in the bowl. <Template which shown as above>

OptionA: Move the right gripper down and then to the left, then forward to grab.

OptionB: Move the right gripper up and then to the left, then forward to grab.

OptionC: Move the right gripper forward and then to the right, then down to grab.

OptionD: Move the right gripper down and then to the right, then forward to grab.

OptionE: Move the right gripper down and then to the left, then backward to grab.

Figure 23: Example of *Execution* from the AgiWorld dataset. The model must select the correct primitive action, expressed in normalized directional terms, to complete the manipulation goal.

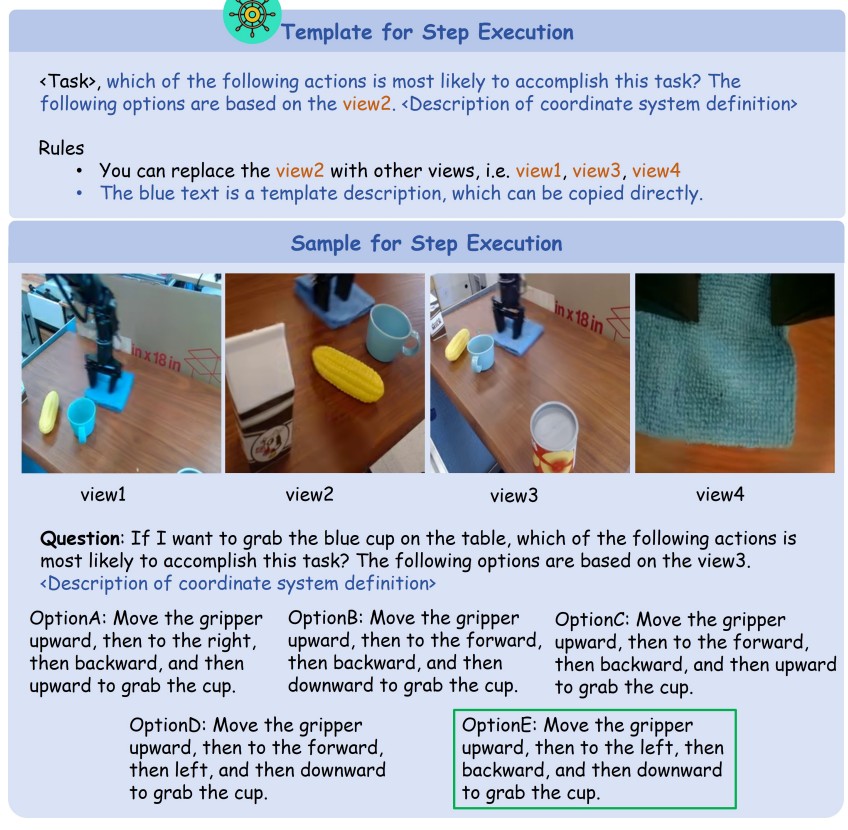

Figure 24: Example of *Execution* from the BridgeV2 dataset. One reference view (here, `view3`) is used to describe the options, and the model must identify the correct low-level action sequence to accomplish the given task.

**📍 Template for Trajectory Selection**

You are given three synchronized views—the left-gripper camera, the head camera, and the right-gripper camera. `<Task>`, which color track is most likely to complete the task?

Rules
- The blue text is a template description, which can be copied directly.

**Sample of Trajectory Selection**

left gripper view          head view          right gripper view

**Question**: You are given three synchronized views—the left gripper camera, the head camera, and the right gripper camera. If I want to place the pink wet wipes into the plastic bag, which color trajectory is most likely to accomplish this task?

OptionA: The blue line.

OptionB: The green line.

OptionC: The pink line.

OptionD: The red line.

OptionE: The yellow line.

Figure 25: Example of *Trajectory Evaluation* from the AgiWorld dataset. The model must select the correct colored trajectory that successfully completes the described manipulation task.

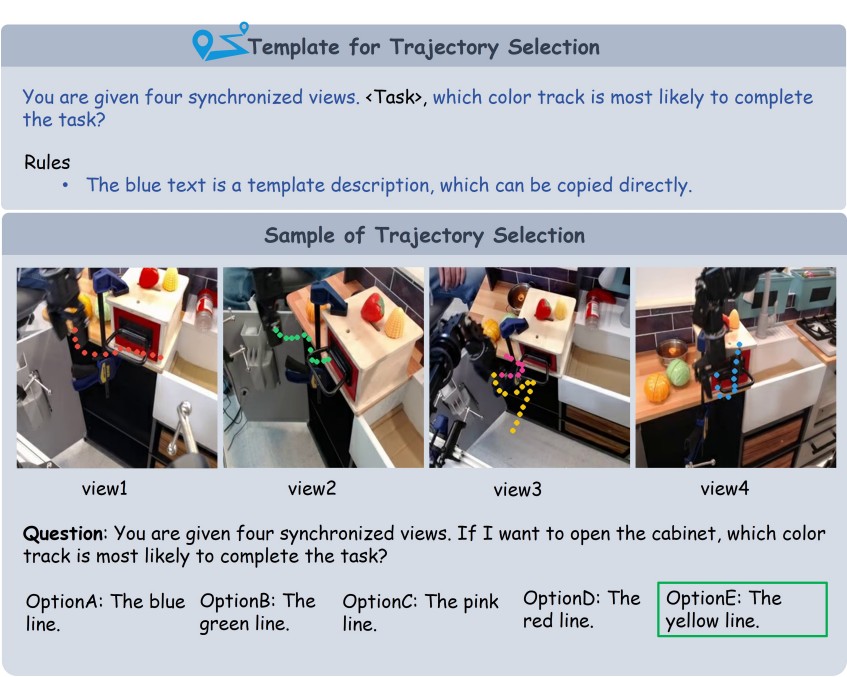

Figure 26: Example of *Trajectory Evaluation* from the BridgeV2 dataset. The model is given four synchronized views and must select the correct colored trajectory (here, the green line) that completes the described task.

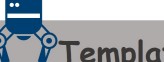**Template for Affordance Recognition**

You are given three synchronized views—the left-gripper camera, the head camera, and the right-gripper camera. Across these images, five grasp candidates are illustrated with color-coded lines—red, yellow, green, blue, and pink—such that each color appears exactly once, on one view only. The two endpoints of every line mark the intended positions of the gripper fingers. Which colour-coded line represents the grasp candidate most likely to succeed?

Rules
- The blue text is a template description, which can be copied directly.

**Sample of Affordance Recognition**

**Question**: As shown above

OptionA: Blue line.     OptionB: Green line.     OptionC: Pink line.

OptionD: Red line.     OptionE: Yellow line.

Figure 27: Example of *Affordance Recognition* from the AgiWorld dataset. Five color-coded grasp candidates are illustrated, and the model must select the one most likely to succeed.

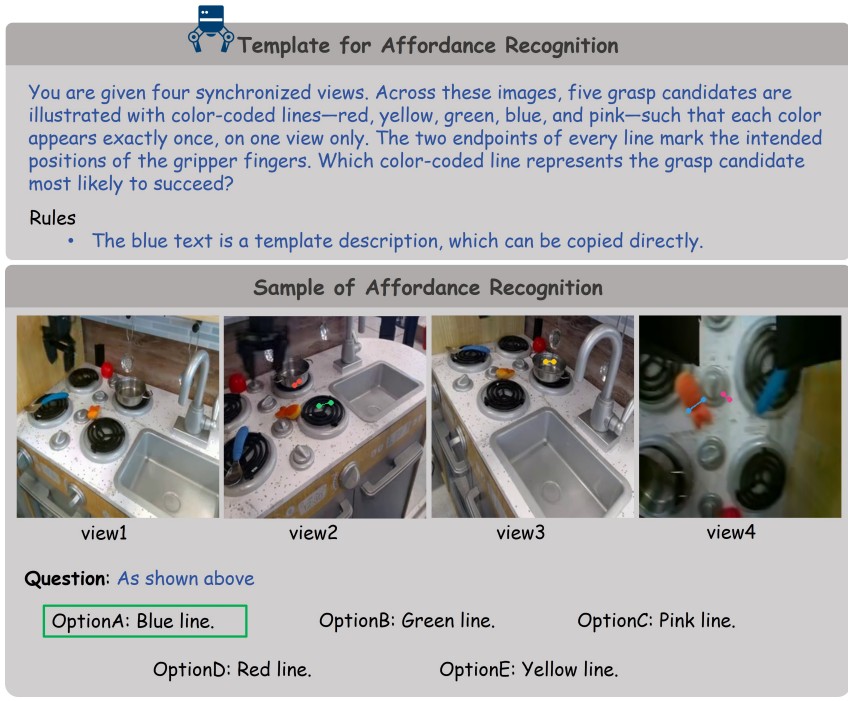

Figure 28: Example of *Affordance Recognition* from the BridgeV2 dataset. Grasp candidates are distributed across four synchronized views, with the model required to select the most feasible option.

## Perceptual Hallucination / Grounding

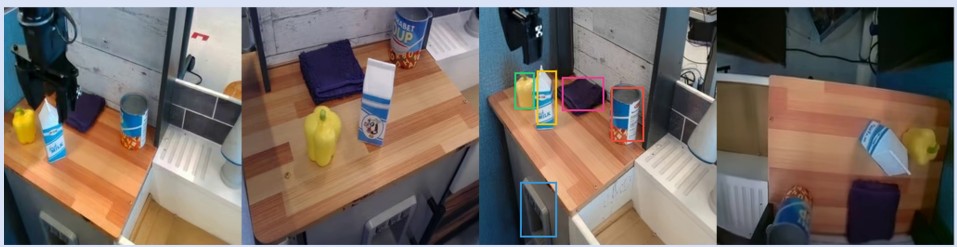

"question": "You are given four synchronized views, only the view3 contains colored bounding boxes. Which option corresponds to the shortest grasping distance?

"A": "Grab the item in the red bounding box."     (Model answer)

"D": "Grab the item in the yellow bounding box." (Correct)

Model response: To determine the shortest grasping distance, we need to consider the relative positions of the objects in the views. The view3 contains colored bounding boxes, which likely indicate the objects of interest for grasping.\n\n1. **Red bounding box**: This box is around the yellow bottle.\n2. **Pink bounding box**: This box is around the purple cloth.\n3. **Green bounding box**: This box is around the blue bottle.\n4. **Yellow bounding box**: This box is around the yellow bottle.\n5. **Blue bounding box**: This box is around the blue bottle.\n\nFrom the images, the yellow bottle is the same object in all views. The red bounding box in view3 is the closest to the yellow bottle in view4, indicating the shortest grasping distance.\n\nAnswer: A

Analyze: It mistook the red soup can for a "yellow bottle" and the green bell pepper for a "blue bottle." This is the most obvious "hallucination," directly proving that the small model failed at the Perception level.

Figure 29: **Case Study 1: Perceptual Hallucination (Qwen2.5-VL-7B).** The model fails at basic visual grounding. As shown in the response, it explicitly misidentifies the red bounding box (a soup can) as a "yellow bottle" and the green box (a pepper) as a "blue bottle." This hallucination leads to a baseless conclusion, illustrating the perception bottleneck in smaller models.

## Instance-level Correspondence

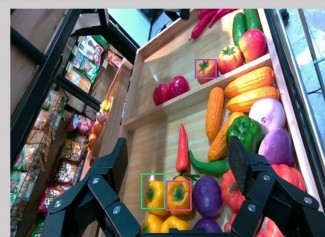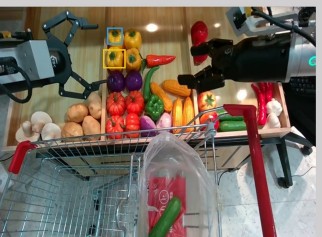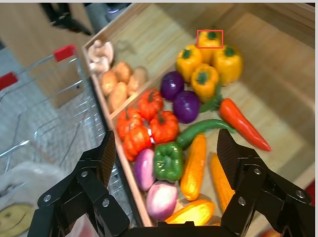

"question": "In the right-gripper camera view, the item is outlined with a red bounding box. Which colored bounding box encloses that same item in the left-gripper camera view and the head camera view?"

"B": "The yellow bounding box." (Model answer)     "E": "The blue bounding box. (Correct)

Model response: To determine which colored bounding box encloses the same item in both the left-gripper camera view and the head camera view as the red bounding box in the right-gripper camera view, let's follow these steps:\n\n1. Identify the item within the red bounding box in the right-gripper camera view. It appears to be a yellow bell pepper.\n2. Locate the yellow bell pepper in the left-gripper camera view. It is enclosed by the pink bounding box.\n3. Confirm the location of the yellow bell pepper in the head camera view. It is enclosed by the yellow bounding box.\n\nGiven this analysis, the item outlined by the red bounding box in the right-gripper camera view corresponds to the pink bounding box in the left-gripper camera view and the yellow bounding box in the head camera view.\n\nSince the question asks for the color of the bounding box that encloses the same item in both the left-gripper camera view and the head camera view, we need to identify the common item and its corresponding bounding box colors. The yellow bell pepper is consistently identified by the yellow bounding box in the head camera view.\n\nTherefore, the correct answer is:\n\nAnswer: B

Analyze: The main problem here is that the model first identifies what the object is when making its judgment. However, in this scene, there are many identical objects. The model must correctly identify the correct object based on spatial location and multi-view consistency. This turns it into a simple object recognition problem.

Figure 30: **Case Study 2: Instance-Level Correspondence Failure (Qwen2.5-VL-72B).** The scene contains multiple instances of the same class (yellow peppers). The model correctly identifies the category but fails to perform geometric alignment. It selects Option B solely because it contains a "yellow bell pepper," effectively reducing the multi-view consistency task to a simple object recognition problem.

## Frame-of-reference / Single-view Bias

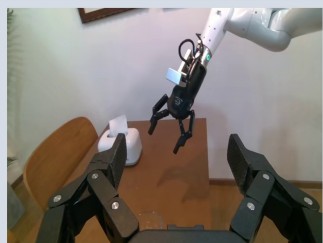 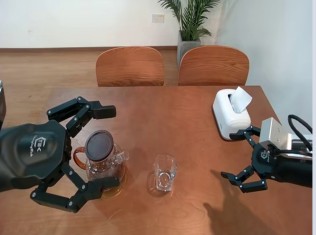 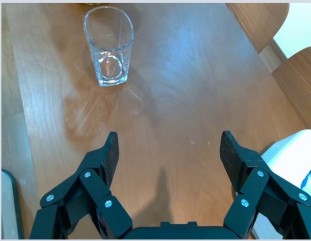

"question": "If I want to pour water into the cup, which of the following operations is most likely to complete the task with the least collision? The following options are based on the left gripper camera view. <The definition of Coordinate>

"A": "Move the left gripper forward, then left, and then downward to grasp the handle of the kettle. Then move upward, and then forward to pour water into the cup.", (Model Answer)

"E": "Move the left gripper backward, then left, and then downward to grasp the handle of the kettle. Then move upward, and then forward to pour water into the cup." (Correct)

Model response: Step 1: Reasoning\n- Goal: Grasp the kettle handle with the left gripper and then pour into the cup with minimal collision.\n- From the head view, the kettle is on the left side with its handle on the far-left; the cup is near the center. From the left-gripper view, the right gripper/arm is ahead and slightly to the right, so moving right increases collision risk, while moving left avoids it.\n- Safe approach to the handle:\n  - Move forward (+y) toward the kettle.\n  - Then move left (-x) to align with the handle on the kettle's left side, steering away from the right arm.\n  - Then move downward (-z) to grasp the handle near the table surface.\n- After grasping, lifting upward (+z) and moving forward (+y) toward the cup positions the kettle to pour.\n\nWhy others are less suitable:\n- B: Moving right heads toward the right gripper and tissue box; "upward to grasp" is incorrect since the handle is lower.\n- C: Downward then upward before grasping is unnecessary and risky near the table; lacks lateral alignment with the handle.\n- D: Moving right first increases collision risk with the right gripper; the right-then-left path is inefficient.\n- E: Moving backward moves away from the kettle, not toward it; adds unnecessary motion.\n\nStep 2: Best option\nAnswer: A

Analyze: GPT-5 interprets "moving forward" as reaching for the cup (based on the intuition of the Head View), but in the Left Gripper coordinate system, "moving forward" actually means passing over the cup, and going any further down would make it impossible to grab it. This perfectly explains "reference frame confusion" and "single-view offset".

Figure 31: **Case Study 3: Frame-of-Reference Error (GPT-5).** The task requires planning in the *Left-Gripper* frame. The model's reasoning ("Move forward ... toward the kettle") reveals a reliance on the global *Head View* intuition. In the specific local frame of the left gripper, "Forward" actually leads to a collision with the cup, while the correct action is to move "Backward" (Option E).

