# OpenReview forum: "Seeing Across Views: Benchmarking Spatial Reasoning of Vision-Language Models in Robotic Scenes"
_ICLR.cc/2026/Conference — ICLR 2026 Poster_

### Official Review · Reviewer_Awkn · 2025-10-31

**Soundness:** 3
**Presentation:** 3
**Contribution:** 3
**Rating:** 8
**Confidence:** 4

**Summary:**

This paper introduces a novel and valuable benchmark called MV-RoboBench, specifically engineered to test the multi-view spatial reasoning capabilities of Vision-Language Models (VLMs) within the context of robotic manipulation. Diverse existing VLMs, along with enhanced versions incorporating CoT-inspired techniques, are evaluated on the benchmark.

**Strengths:**

1. The benchmark advances beyond traditional multi-view reasoning tasks by integrating them with explicit robotic execution tasks. The accompanying analysis establishes a strong, positive correlation between spatial intelligence and robotic task execution in these multi-view scenarios, validating the design and highlighting the necessity of integrated capability.
2. MV-RoboBench consists of 1.7k manually curated QA items across eight carefully defined subtasks, ensuring comprehensive coverage and a balanced dataset. The construction process utilized a detailed human-in-the-loop quality review pipeline, including rigorous checks for question-answer alignment and randomization of answer options to prevent model bias.
3. A diverse array of existing state-of-the-art models was thoroughly evaluated, including both open-source and closed-source VLMs, as well as variants incorporating CoT-inspired enhancements. This broad evaluation provides a robust baseline against which models are shown to remain far below human performance.
4. The study includes rich analytical results that explore the relationship between perception and action. A particularly insightful finding is that strong performance on general-purpose single-view spatial benchmarks does not reliably transfer to the demanding, embodied tasks within MV-RoboBench, underscoring the unique challenge of multi-view reasoning in robotics.

**Weaknesses:**

1. A typo exists in Figure 3, where "AgiWolrd" should be corrected to "AgiWorld".
2. The paper lacks transparency regarding the human effort involved in its construction. It should expose the origin of the hired annotators, the total number of human-hours dedicated to selecting image pairs and building the 1.7K QA items, and the compensation structure for the annotators.
3. Since the benchmark is grounded in robotic manipulation (the domain of VLA models) and utilizes real robotic demonstration data, a VLA-centric evaluation would be highly beneficial, as VLAs are specifically designed for the physical execution and grounding that this benchmark demands, contrasting sharply with the training data distribution of many general-purpose VLMs.

**Questions:**

1. Given that Action Planning and Step Execution answers rely on normalized directional terms (e.g., leftward, forward), how were these linguistic action sequences generated and validated to be reliable and unbiased? Were these descriptions manually authored by annotators based on visual observation, or were they extracted directly from existing action annotations in the source datasets? What specific measures were implemented to ensure the reliability and eliminate ambiguity or bias in the normalized directional language, especially concerning the coordinate system definition and the consistency of the distractor options?
2. About Trajectory Selection questions, what is the origin of the ground-truth trajectories (e.g., raw robot waypoints from the AgiWorld/BridgeV2 data vs. human-drawn ideal paths)? What methodology was used to create the distractor options to ensure they are visually plausible but fail due to collision or poor geometric alignment, thus demanding multi-view reasoning?

---

> ### Author Response · Authors · 2025-11-22
> **Response to Reviewer Awkn - Weakness 1 and 2**
>
> Dear Reviewer,
>
> **For weakness 1:** We thank the reviewer for the careful reading. We have corrected the typo “AgiWolrd” to “AgiWorld” in Figure 3 in the revised manuscript.
>
> ---
>
> **For weakness2:** We appreciate the reviewer’s request for more transparency about how MV-RoboBench was constructed. In the revised manuscript we have added a dedicated section in **Appendix E** (“Human Annotation Protocol and Quality Assurance”, Sec. E.5) that explicitly documents the annotators’ background, the human-hour estimates, and the quality-control procedures.
>
> Concretely:
> - **Annotator origin.** All annotators are **senior undergraduate students or Ph.D. candidates in computer science or closely related fields** from our institution. They are not crowd workers or external contractors, but trained student collaborators with prior exposure to robotics or vision–language research.
> - **Human-hour estimates.** We now report the total effort broken down by stage (Appendix E.5.2):
>   – **Collection and filtering of image pairs:** ~**200** hours (dataset selection, automatic pre-filtering scripts, plus manual screening of ~3K candidate multi-view pairs).
>   – **QA construction and iterative refinement:** ~**600** hours (defining subtasks, training annotators, and manually authoring / revising 1.7K QA items, including bounding boxes, cube configurations, and distractor design).
>   – **Cross-checking and validation:** ~**400** hours (multi-annotator review, discussion of flagged items, and revision or discard of problematic questions).
>   In total, this amounts to roughly **1,200 human hours** devoted to building and validating the benchmark.
> - **Compensation.** Since all annotators are students within our research group, their participation was part of ongoing research projects and did **not** involve separate per-item monetary compensation; there is therefore no external payment scheme or crowd-sourcing platform involved.
>
> In addition, Appendix E.5.1–E.5.3 now describe our **multi-stage annotator training**, the use of **standardized tools** (Qt-based multi-view label interface and 3D cube tool), and a **multi-round human-in-the-loop quality-control pipeline** (including hard distractor design and multi-annotator cross-validation). We hope these additions make the construction process and human effort substantially clearer and address the reviewer’s concern about transparency.
>
> Best regards,

---

> ### Author Response · Authors · 2025-11-22
> **Response to Reviewer Awkn - Weakness 3**
>
> Dear reviewer,
>
> We thank the reviewer for this very insightful comment and we fully agree that a VLA-centric evaluation is a highly desirable *ultimate* goal for a benchmark grounded in real robotic manipulation data. Our long-term vision is indeed to evaluate full VLA systems on top of MV-RoboBench; in this first step, however, we focus on a more fundamental question: **how well can models reason about multi-view robotic scenes at the perception–cognition level, before any low-level control is involved?**
>
> Conceptually, we view an embodied VLA failure as the result of two entangled factors:
> - **High-level reasoning errors (“brain”)** – wrong spatial understanding, faulty planning, incorrect affordance judgement.
> - **Low-level execution errors (“hands”)** – controller instability, dynamics/modeling error, actuator limits, environment resets, etc.
>
> End-to-end VLA roll-outs inevitably mix these two sources of error, which makes it hard to tell whether a failure comes from poor **multi-view spatial reasoning** or from suboptimal **low-level control**. **MV-RoboBench is deliberately positioned as a “unit test” of the reasoning core**: it strips away the “hands” factor and isolates whether a model can interpret synchronized multi-view observations, reason in a well-defined 3D coordinate frame, and choose collision-aware / task-feasible options at the abstract action level. We believe this kind of diagnostic signal is a necessary prerequisite before making strong claims about full VLA behavior.
>
> In addition, there are practical constraints that currently make a clean, large-scale VLA comparison difficult:
> (i) **Cost and scalability.** Systematic VLA evaluation requires robot hardware time, environment resets, and repeated roll-outs for statistical robustness, which would drastically limit the number of models and settings we can compare in a fair way.
> (ii) **Lack of standardized VLM→VLA fine-tuning.** There is not yet a widely accepted pipeline for turning a general-purpose VLM into a VLA policy; performance is highly sensitive to dataset choices, action abstractions, and training hyper-parameters. This introduces substantial bias and fairness issues across models, making it difficult to attribute differences to spatial reasoning ability rather than engineering specifics.
>
> That said, we completely share the reviewer’s perspective that **VLA-centric evaluation is the natural next step built on top of this work**. We will clarify this positioning in the revised manuscript: MV-RoboBench is intended as a **diagnostic benchmark for the multi-view spatial “brain”** of embodied systems, and we are actively exploring how to extend it to controlled VLA roll-outs once the above confounding factors (cost and fine-tuning variability) can be better standardized. We hope this makes our current scope and future plan clearer.
>
> Best regards,

---

> ### Author Response · Authors · 2025-11-22
> **Response to Reviewer Awkn - Question 1**
>
> Dear reviewer,
>
> Thank you for this careful question about the reliability of the normalized directional descriptions.
>
> 1. **Source of the action sequences.**
>    All directional sequences (e.g., “move leftward, then forward”) are **manually authored by trained annotators from the synchronized multi-view images**, rather than extracted from low-level action logs. The original datasets mainly provide numeric control signals (joint states, end-effector poses), which do not directly encode the high-level semantic plans we aim to evaluate. We therefore explicitly ask annotators to “think like a VLM/VLA brain” and describe the intended motion in normalized language.
>
> 2. **Coordinate system and ambiguity control.**
>    To avoid ambiguity, all terms are anchored to the **camera-centric right-handed coordinate system** defined in Appendix E.3 (gravity → up/down, camera optical axis → forward/backward, right-hand rule → left/right). Annotators are restricted to this vocabulary: *forward/backward, leftward/rightward, upward/downward*. During construction, we **discard examples where the dominant direction is unclear** (e.g., near-diagonal motions that could plausibly be both “forward” and “leftward”), so only axis-dominant actions are retained.
>
> 3. **Distractor construction and validation.**
>    Distractors are created under the **same coordinate system and vocabulary** as the ground-truth sequence and are designed to be **semantically plausible but geometrically incorrect**, while keeping the wording style identical. Multiple annotators then cross-check each item; if any annotator believes that more than one option could be interpreted as correct, the item is revised or removed. We also perform an automatic check: after canonicalizing and sorting the directional tokens, **no two options are allowed to be identical**. Only questions with a unique, unambiguous correct sequence are kept.
>
> We have added these clarifications on the generation, normalization, and validation of directional language to Appendix E in the revised version.
>
> Best regards,

---

> ### Author Response · Authors · 2025-11-22
> **Response to Reviewer Awkn - Question 2**
>
> Dear reviewer,
>
> We appreciate the reviewer’s question about the origin and difficulty design of the Trajectory Selection items.
>
> (1) Ground-truth trajectories: human-drawn “ideal” paths.
> For this subtask we do **not** use raw waypoints from AgiWorld/BridgeV2 as ground truth. Instead, trained annotators draw **smoothed, idealized trajectories** directly on top of synchronized multi-view frames using our custom Qt-based annotation tool (Appendix E.1). The annotators first inspect the underlying demonstration to understand the intended motion (e.g., approach–grasp–place), and then sketch a **collision-free, geometrically consistent path** that clearly reaches the target in all views. Candidate items where the trajectory just grazes obstacles, is hard to interpret in some view, or is not clearly executable are simply discarded. This makes the ground-truth path both realistic (aligned with the original demo) and visually unambiguous for evaluation.
>
> (2) Distractor trajectories: visually plausible, geometrically wrong, with a difficulty ladder.
> Distractors are generated by **controlled perturbations** of the ground-truth path, rather than arbitrary scribbles:
>
> - We include a small number of **“easy” negatives** that clearly violate physical constraints (e.g., cutting through solid objects or going well past the target). These help filter out models that cannot perform basic collision reasoning.
> - More importantly, each item also contains **1–2 hard negatives** that are **smooth and intentional in a single view**, but become invalid when all views are considered—for example, a path that goes around the wrong side of an obstacle or stops slightly off the target when projected into depth. These are created by locally shifting segments of the ground-truth path or misaligning key waypoints, so that the curve still “looks reasonable” but is **geometrically inconsistent in 3D**.
>
> As part of the human-in-the-loop quality control described in Appendix E.5(Quality-Control Procedures), multiple annotators independently review each question. If any distractor could reasonably be interpreted as collision-free and correctly aligned across all views—or if the difference from the ground truth is not clearly attributable to spatial/geometric reasoning—the item is revised or removed. This layered design ensures that (i) the ground-truth trajectories are genuinely executable in the scene, and (ii) the distractors are visually plausible yet geometrically wrong, so that solving the task truly requires multi-view spatial reasoning rather than spotting obviously bad curves.
>
> Best regards,

---

> > ### Comment · Reviewer_Awkn · 2025-11-24
> >
> > Thanks for the newly added explanations and clarifications. These revisions make the paper more complete. In particular, the clarified positioning is more concise and presents the contribution in a clearer, more reasonable manner. The updates address my earlier concerns, and I will keep my original score of 8 (accept, good paper).

---

> > > ### Author Response · Authors · 2025-11-24
> > > **Thanks for your valuable feedback**
> > >
> > > We sincerely thank Reviewer Awkn for the positive feedback and for acknowledging the improvements in our revisions. We are glad to hear that the new experiments and clarifications have successfully addressed your concerns. We will incorporate the corresponding results and explanations into the revised manuscript so that the improvements are fully reflected in the final version. Thank you again for your insightful and constructive suggestions—they have been truly valuable for strengthening our work.

---

### Official Review · Reviewer_drdc · 2025-11-01

**Soundness:** 3
**Presentation:** 4
**Contribution:** 3
**Rating:** 6
**Confidence:** 5

**Summary:**

This work introduces MV-RoboBench, a benchmark specifically designed to evaluate the multi-view spatial reasoning capabilities of vision-language models (VLMs) in robotic manipulation. The benchmark contains 1.7k manually curated QA items covering eight subtasks. Evaluation results show that state-of-the-art models still perform far below human levels. The study also explores chain-of-thought (CoT)-inspired enhancements, which produce mixed, model-dependent effects and reveal two main insights:
(1) spatial reasoning ability strongly correlates with robotic execution performance; and
(2) high accuracy on general single-view spatial benchmarks does not reliably transfer to multi-view robotic tasks.

**Strengths:**

1. The proposed benchmark addresses an important and practical problem in robotics.
2. The annotations for the benchmark are manually collected, which helps ensure their correctness and reliability.
3. This work also investigates the effects of CoT prompting and uncovers two key correlations, between spatial reasoning and robotic execution, and between single- and multi-view understanding, offering interesting insights to the research community.
4. The presentation is easy to follow.

**Weaknesses:**

1. The paper claims to be the first to integrate spatial and robotic reasoning with synchronized multi-view inputs in robotic manipulation scenarios. However, the previous ERQA benchmark also includes some multi-view spatial reasoning and manipulation questions in its test set. Although such samples are fewer and many ERQA items are single-image based, it nevertheless contains similar tasks, such as cross-view matching, as those in the proposed benchmark. Therefore, the authors should discuss ERQA in the paper.
In addition, another benchmark, MMSI-Bench, also focuses on multi-image spatial understanding. Although it is not primarily designed for manipulation scenarios, it contains samples from datasets like Agibots, which are relevant. The authors should be aware of and discuss this work as well, to provide readers with a more comprehensive understanding of the existing benchmark landscape in this domain.
2. Although the answers in the proposed benchmark are manually crafted, the questions are generated using template-based designs, which may limit the diversity and richness of the dataset.
3. While the benchmark emphasizes multi-view understanding, the paper does not provide a mechanism to ensure that questions cannot be answered by observing only a single image. For example, in Figure 1, the distance judgment and spatial consistency examples appear solvable from one view alone. I recommend conducting another round of quality review to verify each question’s validity and to filter out those that can be answered using only a single image.
4. Regarding the experimental results in Figure 6, the authors use the OmniSpatial benchmark. However, I do not believe OmniSpatial is an appropriate choice for this analysis, as it includes many samples with multiple images, not-real scenes, and non-robotic content. This may distort the analysis of the correlation. A more suitable comparison would be to use the single-view questions from the ERQA benchmark instead.
5. This work did not have an error analysis of the tested models.

I am open to raising the score if my concerns are addressed.

**Questions:**

1. The model performance on the 3D spatial consistency subset appears quite unusual. In particular, GPT-5-Chat achieves only 4.9%, whereas GPT-5-Mini reaches 72.55%. I am curious about the API identifiers of these two models. This large discrepancy also suggests that there might be an evaluation bug. It would be helpful if the authors could provide some example predictions from these models to illustrate and better understand the observed performance gap.

2. I also wonder whether the authors believe there are potential methods to automatically collect such QA pairs for training purposes. If so, I am curious how fine-tuned models would perform on this benchmark. Moreover, the limited chain-of-thought (CoT) effects observed in this work might be due to the fact that the models have not been exposed to such reasoning patterns during training. Fine-tuning with CoT-style data could potentially enhance their performance further.

---

> ### Author Response · Authors · 2025-11-22
> **Response to Reviewer drdc - Weakness 1**
>
> Dear reviewer,
>
> We thank the reviewer for pointing out ERQA and MMSI-Bench. We agree that explicitly discussing them improves the positioning of our benchmark.
>
> In the revised manuscript, we (i) **added ERQA and MMSI-Bench to Table 1** with their “Multi-View” status marked as **Partial**, and (ii) **extended the related-work discussion** below Table 1 to describe both benchmarks and refine our novelty claim.
>
> **Relation to ERQA.**
> We now explicitly acknowledge that ERQA has already made valuable early explorations on multi-image and manipulation-related spatial reasoning. Building on this line of work, we clarify two key differences:
>
> - **Coverage and depth of multi-view robotic data.** ERQA has about **0.4K** QA pairs, with only **≈0.1K** being multi-image and an even smaller subset truly synchronized multi-view in robotic scenes. In contrast, **all 1.7K QA items in MV-RoboBench are built from synchronized multi-camera robot demonstrations**, and every question is intentionally tied to this multi-view setting. Moreover, ERQA’s multi-view items are relatively basic (often object-level consistency), whereas MV-RoboBench also includes **3D spatial subtasks** (spatial cube reasoning, viewpoint identification, distance under occlusion) and **robotic reasoning subtasks** (action planning, step execution, trajectory and affordance selection).
>
> - **Robot-centric design.** ERQA spans heterogeneous embodied setups (human egocentric, static scenes, some robot contexts). MV-RoboBench is **fully grounded in robotic manipulation**, with a fixed, synchronized head + gripper camera rig, which allows us to systematically study how multi-view spatial understanding supports **robotic task reasoning** within a single, coherent manipulation environment.
>
> **Relation to MMSI-Bench.**
> We also describe MMSI-Bench in the revised Table 1 as a benchmark for **multi-image spatial understanding** across diverse domains (including some Agibots data), whose questions mainly probe generic multi-image perception. In contrast, **MV-RoboBench is fully grounded in multi-view robotic manipulation scenes and is explicitly designed to stress (i) fine-grained consistency across synchronized views, (ii) holistic 3D understanding of the workspace, and (iii) using this spatial state as a “robotic brain” to support action-centric decisions**.
>
>
> Overall, we have revised the paper to present ERQA and MMSI-Bench as **important but complementary** efforts: they introduced multi-image and embodied spatial reasoning, while **MV-RoboBench systematically extends this direction to a larger-scale, synchronized multi-view benchmark with richer 3D spatial and robotic reasoning subtasks in real robot manipulation scenes**
>
> Best regards,

---

> ### Author Response · Authors · 2025-11-22
> **Response to Reviewer drdc - Weakness 2**
>
> Dear reviewer,
>
> We appreciate the reviewer’s concern about potential limitations of template-based question design, and we agree that **diversity and richness are important for a useful benchmark**.
>
> In designing the multi-view spatial subtasks (**Cross-View Matching, Viewpoint Identification, 3D Spatial Consistency**), our intention was to primarily stress the models’ ability to **exploit synchronized multi-view geometry**—such as cross-view consistency, viewpoint changes, and 3D layout—rather than their ability to handle a wide variety of surface linguistic forms. For these subtasks, we therefore **keep the linguistic patterns relatively simple and standardized, while letting the diversity come from the visual and geometric side**: MV-RoboBench is constructed from on the order of ~1K real robot demonstration episodes across diverse tasks and scenes in AgiWorld and BridgeV2, leading to a wide range of object configurations, occlusions, camera poses, and multi-view relationships. Even under a shared template, the required reasoning differs substantially from instance to instance because the **underlying 3D geometry and viewpoints are always changing**.
>
> For the remaining subtasks, the questions are **not produced by blindly instantiating a single fixed template**. As detailed in the annotation protocol (Appendix F), annotators start from concrete multi-view scenes and **manually design high-level actions and distractors tailored to each situation** (e.g., which object should be manipulated, which trajectory is prone to collision, which affordance is feasible). We still use relatively standardized phrasing (e.g., normalized directional terms) as a scaffold to **reduce ambiguity**, but this is a deliberate choice: in practical robotic pipelines, free-form user commands are typically parsed into **structured, canonical motion primitives** before execution. Our benchmark follows the same principle—keeping language controlled and aligned with a well-defined coordinate system—so that **(i) each item has a unique, unambiguous ground-truth answer, while (ii) the richness comes from the diversity of multi-view embodied scenes and task configurations rather than from superficial linguistic variation**.
>
> Best regards,

---

> ### Author Response · Authors · 2025-11-22
> **Response to Reviewer drdc - Weakness 3**
>
> Dear reviewer,
>
> We thank the reviewer for this important point. We agree that, for a benchmark emphasizing multi-view reasoning, it is crucial to show that questions are not trivially solvable from a single image.
>
> Beyond the multi-round human quality control we already applied during construction (multi-annotator review and removal of items where a single view clearly sufficed), we have now added an explicit **single-view vs. multi-view ablation** in the revised manuscript (new Appendix G). For the four subtasks where a single-view setting is still well-posed (*Distance Judgment, 3D Spatial Consistency, Action Planning, Step Execution*), we compare models that see **only the head/center view** with the same models given **all synchronized views**.
>
> Below we report the single-view accuracy and the performance gap Δ relative to the multi-view variant (Δ = multi-view – single-view, in percentage points). Each subtask cell shows `single-view accuracy (+Δ)`:
>
> ```markdown
> | Model          | Avg (MV) | Δ Avg | Dist. Judg.         | 3D Spatial Cons.       | Action Plan.        | Step Exec.        |
> |----------------|----------|-------|---------------------|------------------------|---------------------|-------------------|
> | GPT-5          | 64.65    | +6.69 | 36.32 (+18.90)      | 78.43 (+3.92)          | 76.96 (+2.45)       | 66.24 (+2.14)     |
> | GPT-5-chat     | 28.11    | +3.05 | 29.35 (+13.44)      | 10.29 (-5.39)          | 35.29 (+1.47)       | 36.32 (+3.85)     |
> | GPT-4.1        | 24.20    | +3.59 | 32.84 (+10.44)      | 4.90 (+1.47)           | 30.39 (-0.49)       | 28.21 (+3.41)     |
> | GPT-4o         | 26.81    | +0.88 | 31.84 (+5.47)       | 5.39 (+0.98)           | 33.82 (-0.49)       | 35.04 (-1.28)     |
> | Qwen2.5-VL-72B | 23.01    | +0.87 | 30.85 (+3.98)       | 3.92 (+0.98)           | 30.88 (-2.45)       | 26.07 (+1.28)     |
> | Qwen2.5-VL-32B | 20.28    | -0.07 | 24.38 (+1.49)       | 8.33 (+2.45)           | 25.49 (-1.98)       | 22.65 (-2.99)     |
> | Qwen2.5-VL-7B  | 17.91    | +1.55 | 20.40 (+0.00)       | 4.90 (+3.92)           | 21.08 (+1.47)       | 24.36 (+1.71)     |
> ````
>
> (Full results are provided in Appendix G of the revised paper; the anomalous drop of GPT-5-chat on 3D Spatial Consistency is analyzed separately in our response to R3 (Q1).)
>
> Two points are worth highlighting:
>
> * **For depth-sensitive tasks, multi-view gives a large relative gain.**
>   On *Distance Judgment*—the example highlighted by the reviewer—GPT-5 and GPT-4.1 improve by **+18.90** and **+10.44** points, respectively, when moving from single-view to multi-view. Given the 5-choice format, this corresponds to a large relative improvement over the single-view baseline and confirms that a single image often under-constrains depth and occlusion for current models.
>
> * **Multi-view gains are most visible in strong models, while weaker models hit a floor.**
>   Strong models consistently benefit from multi-view input, especially on spatially demanding subtasks. In contrast, small models sometimes show Δ ≈ 0 or even negative Δ, indicating that when a model cannot fuse views reliably, simply adding more images may hurt. This suggests that MV-RoboBench is indeed probing **non-trivial multi-view fusion**, rather than giving free gains for more pixels.
>
> Taken together with our existing human review process, this ablation serves as a quantitative validation that MV-RoboBench is not “single-view in disguise”: for capable models, performance with only one view is systematically and substantially lower than with full multi-view input, even in examples that may look solvable from a single image with strong monocular priors.
>
> Best regards,

---

> ### Author Response · Authors · 2025-11-22
> **Response to Reviewer drdc - Weakness 4**
>
> Dear reviewer,
>
> We thank the reviewer for raising this point. In Figure 6, our goal is to study a **general-to-specific transfer** question: does strong performance on a **single-view spatial benchmark** predict performance on **MV-RoboBench**, which specifically targets **multi-view, robotic, action-centric spatial reasoning**?
>
> In this context, we use OmniSpatial as a proxy for general single-view spatial ability. While it does include synthetic and non-robotic scenes, we view this as a strength rather than a limitation: the mixture of real and non-real content forces models to handle abstract spatial structure, giving a more comprehensive measure of their single-view spatial intelligence.
>
>
>
> | Model            | ERQA Avg. |
> |------------------|-----------|
> | GPT-5            | 59.34     |
> | GPT-5-mini       | 54.00     |
> | GPT-5-chat       | 49.50     |
> | GPT-5-nano       | 44.00     |
> | GPT-4.1          | 49.00     |
> | GPT-4o           | 46.00     |
> | GPT-4.1-mini     | 46.00     |
> | GPT-4.1-nano     | 38.25     |
> | Qwen2.5-VL-72B   | 44.61     |
> | Qwen2.5-VL-32B   | 44.75     |
> | Qwen2.5-VL-7B    | 43.11     |
>
> As discussed in Appendix D, ERQA is clearly relevant to embodied reasoning and already contains action-related and some multi-view questions. At the same time, the numbers above highlight some limitations when using it as the main axis for our correlation analysis:
>
> 1. **Limited discriminative power for current models.**
>    Most models are compressed into a relatively narrow band (roughly 40–50%). For instance, Qwen2.5-VL-7B at 43.11% and GPT-4o at 46.00% differ by less than 3 percentage points. This small spread makes it difficult to obtain a stable and informative correlation with MV-RoboBench: small fluctuations or noise can dominate any trend we try to fit.
>
> 2. **Strong ERQA performance still does not imply strong MV-RoboBench performance.**
>    Even for models that perform competitively on ERQA, their scores on MV-RoboBench can differ substantially, and the ranking of models is not preserved. In other words, doing well on ERQA does not reliably predict doing well on our multi-view robotic benchmark, which confirms our main message that **existing spatial benchmarks, even domain-related ones like ERQA, do not subsume the capabilities required by MV-RoboBench**.
>
> In the revised version, we (i) report ERQA results in Appendix D and (ii) explicitly discuss ERQA in the main text as a complementary robotics-related benchmark. We continue to use OmniSpatial in Figure 6 because its broader score range makes the correlation analysis more stable, while ERQA provides an additional domain-specific viewpoint. Overall, these additions underline that MV-RoboBench is complementary rather than redundant with existing benchmarks, offering a dedicated evaluation of multi-view robotic spatial reasoning that is not captured by OmniSpatial or ERQA alone.
>
> Best regards,

---

> ### Author Response · Authors · 2025-11-22
> **Response to Reviewer drdc - Weakness 5**
>
> Dear reviewer,
>
> We thank the reviewer for pointing this out. We agree that understanding *how* models fail is important for assessing the value of a benchmark.
>
> In the revised version, we have added a dedicated **error analysis section** (new Appendix J), where we qualitatively analyze failures of three representative models: a strong proprietary model (GPT-5), a strong open model (Qwen2.5-VL-72B), and a weaker open model (Qwen2.5-VL-7B). For each model, we manually inspect 10–20 erroneous cases across core subtasks and summarize the recurring failure modes.
>
> Our analysis reveals several consistent patterns:
>
> - **Single-view bias / weak multi-view fusion:** models often rely on a single “favorite” view (typically the head camera) and ignore contradicting evidence from other synchronized views.
> - **Depth and 3D confusion:** models frequently substitute 2D heuristics (size, screen position) for true 3D reasoning, leading to systematic errors in distance judgment and 3D spatial consistency.
> - **Frame-of-reference and instance-level errors:** even with an explicit camera-centric coordinate system, models mis-handle local frames (e.g., left-gripper vs. head view) and confuse specific object instances when multiple identical objects are present.
> - **Affordance and physical feasibility issues:** in trajectory and affordance tasks, models often favor visually “neat” grasps or paths that would in fact be unstable or collision-prone for a real robot.
>
> These patterns appear not only in weaker models but also in GPT-5, and we illustrate them with concrete examples in Appendix J. This suggests that MV-RoboBench is genuinely diagnostic of systematic weaknesses in multi-view spatial and robotic reasoning, and we plan to gradually include additional representative error cases as we evaluate more models and more cases.
>
> Best regards,

---

> ### Author Response · Authors · 2025-11-22
> **Response to Reviewer drdc - Question 1**
>
> Dear reviewer,
>
> We thank the reviewer for carefully checking the numbers and for raising this concern. We verified that there is **no evaluation or formatting bug** in our pipeline: the very low score of GPT-5-chat on the 3D Spatial Consistency subtask is a genuine outcome. The API identifiers used in our experiments are:
> - GPT-5-chat: `gpt-5-chat_2025-08-07`
>
> To diagnose this discrepancy, we ran an additional experiment where we forced GPT-5-chat to answer with **Chain-of-Thought (CoT)** explanations before selecting an option. Under the original “direct answer” setting, GPT-5-chat achieved only **4.90%** accuracy on 3D Spatial Consistency (far below the 20% random baseline). With CoT, its accuracy on this subtask increased to **30.39%**:
>
> | Setting             | 3D Spatial Consistency |
> |---------------------|------------------------|
> | Direct answer       | 4.90%                 |
> | With CoT (w/ expl.) | 30.39%                |
> | Δ                   | +25.49                |
>
>
> This pattern suggests a systematic misunderstanding of the coordinate system, rather than random noise or a bug. The 3D Spatial Consistency questions rely on the explicit camera-centric right-handed coordinate system defined in Appendix E.3 (gravity + optical axis). In the direct setting, GPT-5-chat appears to default to a flawed internal prior and consistently selects options corresponding to an axis-flipped or misaligned grid. When we ask it to reason step-by-step, it is forced to reread and restate the coordinate definition, which acts as an “axis recalibration” and largely recovers performance on this subtask.
>
> Importantly, this behaviour is not a universal artifact of CoT. For other models where the 3D Spatial Consistency scores are already reasonable (e.g., GPT-4.1, Gemma-3-12B, Qwen2.5-VL-7B), our CoT experiments (Table 3 in the paper) show only small gains, typically within a few percentage points. GPT-5-chat is therefore a special case where CoT reveals and partly corrects a latent coordinate-frame bias, rather than a sign of an evaluation issue.
>
> Best regards,

---

> ### Author Response · Authors · 2025-11-22
> **Response to Reviewer drdc - Question 2**
>
> Dear reviewer,
>
> We appreciate this forward-looking question, which touches exactly on the two dimensions we see as most important for future work: **how to scale data** and **how to better exploit CoT-style reasoning**.
>
> On the data side, we do believe there is substantial potential for more automatic collection of QA pairs for **training** purposes. In MV-RoboBench, we deliberately relied on human-authored questions and answers for **evaluation**, because for precise spatial reasoning, we found that even strong VLMs can hallucinate subtle geometric facts; for a benchmark, this level of risk in the ground truth is unacceptable, so we kept humans in the loop throughout the pipeline (Appendix E). For **training**, however, the constraints are different: it is much more reasonable to combine
> (i) automatic generation from simulation or logged robot states (poses, depth, contact information) and
> (ii) large-scale templating or model-generated QA proposals with human filtering,
> to obtain high-volume but slightly noisy supervision. We see this kind of hybrid, partially automated data construction as a promising way to scale up spatial/robotic reasoning signals beyond what a purely manual benchmark can provide.
>
> Regarding CoT, we fully agree with the reviewer’s hypothesis. Our experiments show that naive CoT prompting has **mixed and model-dependent** effects, and we believe a major reason is that current models have not been explicitly trained on **multi-view geometric CoT traces** of the kind required by MV-RoboBench. In that sense, our CoT results are mainly **diagnostic**: they reveal that the models’ internal reasoning patterns are not yet aligned with the coordinate-system–aware, multi-view–consistent chains of thought that our tasks demand. We share the reviewer’s view that fine-tuning with CoT-style supervision tailored to these tasks—whether generated from simulation metadata, human-written rationales, or a combination—could substantially improve performance, and may even be necessary to “unlock” the full potential of current architectures in this setting.
>
> In summary, we see MV-RoboBench as a first step on the **evaluation** side: it exposes where current models fail in multi-view robotic reasoning. The reviewer’s suggestion points to the natural next step on the **training** side: using similar structures (coordinate systems, multi-view layouts, trajectories) to automatically or semi-automatically generate CoT-augmented training data, and then studying how such fine-tuned models behave on our benchmark. We explicitly highlight this direction as an exciting avenue for future work in the revised version.
>
> Best regards,

---

### Official Review · Reviewer_bybw · 2025-11-02

**Soundness:** 4
**Presentation:** 4
**Contribution:** 3
**Rating:** 6
**Confidence:** 3

**Summary:**

The paper built a benchmark for evaluating the spacial reasoning ability of VLMs under multi-view setting. The experiments analyzed the how spatial and robotic reasoning relate within multi-view manipulation, and whether spatial intelligence measured in single-view settings can transfer to embodied multi-view tasks. The results revealed that the VLMs have some common shortcoming in certain spacial reasoning scenario, provided a potential direction of VLM developments.

**Strengths:**

1. The paper has good structure and easy to follow.
2. The author conducted a large scale experiment on multiple modern VLMs, which makes the experimental results and conclusions convicible.
3. The spacial reasoning ability is critical for the development of VLMs. This paper provided a good example of how to evaluate the VLMs and thus might have broad influence to the community.

**Weaknesses:**

1. In abstract, should not use abbreviation when the the first time "CoT-inspired techniques" appears. Therefore "CoT-inspired techniques" -> "Chain of Thought (CoT)-inspired techniques."
2. Does the image orientation matters? For example, if the head camera view is upside down, can the system still get correct inference? This discussion is necessary to determine the VLMs' generalization on spacial reasoning.
3. The benchmark have multiple-choice questions across eight subtasks, each with exactly one correct answer. However, another important aspect is that whether the system can determine there is no correct choice.

**Questions:**

See weaknesses

---

> ### Author Response · Authors · 2025-11-22
> **Response to Reviewer bybw - Weakness 1 and 2**
>
> Dear Reviewer,
>
> **For weakness 1:** We thank the reviewer for the careful reading and attention to detail. We have adopted this suggestion and updated the abstract to spell out the term on first use as **“Chain-of-Thought (CoT)-inspired techniques”**. This change is reflected in the revised manuscript.
>
> ---
>
> **For weakness 2:** We thank the reviewer for raising this insightful question. Whether current VLMs are robust to non-canonical image orientations is indeed important for assessing their spatial generalization, beyond the specific settings we evaluate in the main experiments.
>
> To study this, we added an **Upside-Down stress test** in the revised manuscript (new Appendix H). In this setting, we vertically flip **all static / third-person views** (e.g., the head camera in AgiWorld and all non-gripper views in BridgeV2) while keeping gripper-mounted views unchanged. As clarified in Appendix E.3, our camera-centric coordinate system is defined via the gravity vector and the camera optical axis, rather than 2D pixel axes. Thus, flipping the images does **not** change the physical scene, the coordinate frames, or the ground-truth answers. This manipulation isolates the effect of image orientation on the models’ reasoning, without altering the underlying tasks.
>
> For a concise overview, the table below summarizes the **average accuracy** under the upside-down setting and the performance change Δ (Upside-Down – Original, in percentage points) for all models:
>
> | Model          | Avg (Upside-Down) | Δ vs. original |
> | -------------- | ----------------- | -------------- |
> | GPT-5          | 37.68             | –18.73         |
> | GPT-5-mini     | 30.79             | –7.49          |
> | GPT-5-nano     | 22.42             | –10.33         |
> | GPT-5-chat     | 28.50             | –3.13          |
> | GPT-4.1        | 26.72             | –4.18          |
> | GPT-4o         | 25.10             | –2.49          |
> | GPT-4.1-mini   | 23.94             | –0.04          |
> | GPT-4.1-nano   | 20.08             | –0.77          |
> | Qwen2.5-VL-72B | 23.33             | –0.96          |
> | Qwen2.5-VL-32B | 22.41             | –0.07          |
> | Qwen2.5-VL-7B  | 19.71             | –1.13          |
>
> (Full per-subtask results are provided in the corresponding table in **Appendix H** of the revised paper.)
>
> **Key findings are:**
>
> * **Strong models are highly orientation-sensitive.**
>   The most capable model, GPT-5, suffers a substantial average drop of **–18.73** points. As detailed in Appendix H, this includes large declines on *3D Spatial Consistency* (from 82.35% to 26.47%) and *Distance Judgment* (from 55.22% to 30.35%). GPT-5-mini and GPT-5-nano also drop by –7.49 and –10.33 points, respectively. This indicates that their strong spatial reasoning is learned under canonical “upright” viewpoints during pre-training and does **not** automatically generalize to upside-down inputs.
>
> * **Weaker models exhibit a floor effect.**
>   In contrast, smaller models (e.g., GPT-4.1-mini, Qwen2.5-VL-7B) show almost no change (Δ ≈ 0), largely because their baseline accuracy on spatial tasks is already low. Their behavior appears dominated by pattern matching or near-random guessing, which makes them relatively insensitive to this orientation perturbation.
>
> Overall, these results clarify that current VLMs are **not rotation-invariant** and generalize best within canonical camera poses. At the same time, in practical robotic deployments, camera mounts and perception pipelines are typically engineered to enforce a consistent upright orientation, so this limitation is partly mitigated by system design. We have added this experiment and discussion in Appendix H to make these orientation-related generalization properties explicit, and we appreciate the reviewer’s suggestion that led us to include this analysis.
>
> Best regards,

---

> ### Author Response · Authors · 2025-11-22
> **Response to Reviewer bybw - Weakness 3**
>
> Dear Reviewer,
>
> We thank the reviewer for bringing up this important aspect. We fully agree that the ability to determine that **no option is correct** is crucial for both reliability and safety, and is complementary to selecting the correct option among valid candidates.
>
> In our original benchmark design, we had already partially incorporated this ability: in the **Cross-View Object Matching** subtask, approximately **25%** of the questions have a ground-truth choice of “None of the bounding boxes is correct”, specifically to discourage models from relying only on elimination. Motivated by the reviewer’s comment, we have now made this dimension more explicit and much more challenging via an additional stress test.
>
> **New “None of the above” stress test (Appendix I).**
> In the revised manuscript, we add a dedicated experiment that evaluates rejection capability across **7 of the 8 subtasks** (all except Distance Judgment, where removing the true “shortest” candidate would make the second-shortest candidate trivially correct, rather than yielding a clean “no correct answer” setting). Concretely, for each question we:
>
> * remove the original correct option,
> * insert a new option “None of the above” (always as option E, to avoid positional bias),
> * and treat “None of the above” as the only valid answer.
>
> Thus, solving these questions requires the model to **reject all visually plausible candidates** and explicitly select the rejection option.
>
> For a concise overview, the table below reports the **average rejection accuracy** on this modified dataset (“Reject. Avg”) and the corresponding performance drop (“Drop”, in percentage points) compared to the original benchmark:
>
> | Model          | Reject. Avg |   Drop |
> | -------------- | ----------: | -----: |
> | GPT-5          |       13.43 | -43.29 |
> | GPT-5-mini     |        9.57 | -34.71 |
> | GPT-5-chat     |        1.05 | -28.84 |
> | GPT-5-nano     |        8.61 | -24.02 |
> | GPT-4.1        |        0.60 | -27.35 |
> | GPT-4.1-mini   |        3.74 | -19.09 |
> | GPT-4.1-nano   |        0.78 | -18.21 |
> | GPT-4o         |        0.92 | -23.51 |
> | Qwen2.5-VL-72B |        3.39 | -19.81 |
> | Qwen2.5-VL-32B |        0.28 | -21.70 |
> | Qwen2.5-VL-7B  |        0.49 | -20.31 |
>
> (Full per-subtask results are provided in the corresponding table in **Appendix I** of the revised paper.)
>
> **Key observations:**
>
> * **Severe over-compliance across all models.**
>   Even our strongest model, GPT-5, drops from about **56%** average accuracy on the original benchmark to **13.43%** in the “None of the above” setting. Most other models collapse to **near-zero** rejection accuracy. This suggests that current VLMs almost always “force” a choice among the visual candidates and tend to hallucinate a justification, rather than admitting that none of the options is correct.
>
> * **Selection bias over validity checking.**
>   The results indicate that models are biased toward picking a **relatively similar** or visually plausible option, instead of verifying **absolute correctness**. Choosing “None of the above” requires a qualitatively different behavior: the model must check each candidate against an internal standard of plausibility and systematically reject all of them. Our stress test shows that, even for advanced models, this kind of strict validity judgment is still very weak.
>
> We have added this experiment and analysis in Appendix I. In summary, MV-RoboBench not only measures how often models **succeed** on spatial and robotic reasoning, but also exposes an important **safety gap**: a strong tendency toward over-compliance and difficulty in rejecting invalid options. We believe this provides a valuable signal for future work on uncertainty estimation, calibration, and refusal mechanisms in embodied VLMs.
>
> Best regards,

---

> > ### Comment · Reviewer_bybw · 2025-11-26
> >
> > Thanks, the author's reply addresses my concerns.

---

> > > ### Author Response · Authors · 2025-11-26
> > > **Acknowledgement**
> > >
> > > We sincerely thank Reviewer bybw for reviewing our updated response and for confirming that your earlier concerns have been addressed. We appreciate your time and constructive feedback throughout the process.

---

### Official Review · Reviewer_oGrs · 2025-11-03

**Soundness:** 2
**Presentation:** 3
**Contribution:** 2
**Rating:** 6
**Confidence:** 4

**Summary:**

Paper presents a new benchmark for assessing cross-view reasoning-related functionalities of AI models, with a focus on tabletop manipulation scenarios. The benchmark is constructed with a 3-step procedure: 1) collecting multi-view frames (mostly left arm, right arm, and center view) from canonical robotic datasets (AgiBot  World, etc); 2) generating QA pairs by crowdsourcing with human raters and QA templates were adopted; 3) curating the benchmark with human raters to obtain a total of 1.7k QA entries. The dataset covers a broad range of reasoning tasks, primarily categorized into spatial reasoning and robotic task reasoning. The authors have conducted a rich collection of experiments, including general benchmarking on foundation AI models, the effectiveness of CoT-enhanced reasoning, the correlation between the performances of spatial reasoning and robotic task reasoning, and whether performances on single-view reasoning benchmarks could transfer to the proposed multi-view benchmarks.

**Strengths:**

+The paper is clearly written and easy to follow. The topic (benchmarking of AI models, embodied AI, spatial AI) is relevant to a broad spectrum of ICLR community members.

+The proposed benchmark is constructed and curated in a rigorous fashion, where human raters, instead of automatic pipelines, perform most annotation tasks. Compared to related benchmarks, this alleviate the difficulty of data curation and potentially help with a higher dataset quality.

+I appreciate the authors for arranging such rich collection of tasks. Notably, showing the correlation between spatial reasoning and robotic task reasoning, as well as the transferabiliy from single view to multi-view reasoning, could indeed shed some light into future research.

**Weaknesses:**

My primary concerns lies with the validity of the "multi-view" setting in the current dataset format and relation to embodied AI tasks.

-It seems that many of the designed tasks do not nessitate the need for multi-view input. For the tasks illustrated in Figure 1, most tasks seems doable with only one view, except for viewpoint identification and cross-view matching. If this is the case, I would like to see an ablation on using only one view (say the center view) as input and see if there is significant performance drop -- of course the two tasks mentioned above should be excluded.

-Although there are tasks that indirectly evaluate the capability of robot manipulation (i.e., robot task reasoning), but it would be great to see if there is more direct connection, let's say the AI models control the robot directly. The authors claim this in the abstract "spatial intelligence and robotic task execution are correlated in multi-view robotic scenarios". However, the "robotic task" in the proposed dataset never touches controlling an actual robot but merely works on reasoning aspects of high-level planning. If the authors choose to go with the current setting, I suggest modifying their claim into "reasoning for high-level planning", not robot task executation in general. Otherwise, experiments on maybe fine-tuning these models into low-level control should be necessary.

**Questions:**

See "Weaknesses".

---

> ### Author Response · Authors · 2025-11-22
> **Response to Reviewer oGrs – Weakness 1**
>
> Dear Reviewer,
> Thank you for raising this central point. We agree that it is important to show **when multi-view input is actually necessary**. Following your suggestion, we added an explicit single-view vs. multi-view ablation in the revised manuscript (new Appendix G, summarized in Section 3.2).
>
> **Ablation setup.**
> We compare each model in two settings: (i) **single-view**, where the model only sees the head/center view, and (ii) **multi-view**, where it receives all available views.
>
> * **Tasks.** We consider four subtasks that remain well-posed with a single view: *Distance Judgment, 3D Spatial Consistency, Action Planning,* and *Step Execution*. Tasks whose questions are intrinsically cross-view (e.g., *Viewpoint Identification, Cross-View Matching, Trajectory Selection, Affordance Recognition*) are not included, because removing views would make many questions ill-posed rather than easy.
> * **Models.** We reuse the same diverse set of models as in the main benchmark: strong proprietary reasoning models (GPT-5, GPT-5-chat, GPT-4.1, GPT-4o) and open-source VLMs (Qwen2.5-VL-72B/32B/7B), allowing us to study both *whether* multi-view helps and *how* this benefit scales with capability.
>
> **Quantitative results.**
> The table below reports **single-view accuracy** and the **gap Δ** to the corresponding multi-view variant (Δ = multi-view – single-view, in percentage points). Each subtask cell shows
> `single-view accuracy (+Δ)`:
>
> ```markdown
> | Model          | Avg (MV) | Δ Avg | Dist. Judg.         | 3D Spatial Cons.       | Action Plan.        | Step Exec.        |
> |----------------|----------|-------|---------------------|------------------------|---------------------|-------------------|
> | GPT-5          | 64.65    | +6.69 | 36.32 (+18.90)      | 78.43 (+3.92)          | 76.96 (+2.45)       | 66.24 (+2.14)     |
> | GPT-5-chat     | 28.11    | +3.05 | 29.35 (+13.44)      | 10.29 (-5.39)          | 35.29 (+1.47)       | 36.32 (+3.85)     |
> | GPT-4.1        | 24.20    | +3.59 | 32.84 (+10.44)      | 4.90 (+1.47)           | 30.39 (-0.49)       | 28.21 (+3.41)     |
> | GPT-4o         | 26.81    | +0.88 | 31.84 (+5.47)       | 5.39 (+0.98)           | 33.82 (-0.49)       | 35.04 (-1.28)     |
> | Qwen2.5-VL-72B | 23.01    | +0.87 | 30.85 (+3.98)       | 3.92 (+0.98)           | 30.88 (-2.45)       | 26.07 (+1.28)     |
> | Qwen2.5-VL-32B | 20.28    | -0.07 | 24.38 (+1.49)       | 8.33 (+2.45)           | 25.49 (-1.98)       | 22.65 (-2.99)     |
> | Qwen2.5-VL-7B  | 17.91    | +1.55 | 20.40 (+0.00)       | 4.90 (+3.92)           | 21.08 (+1.47)       | 24.36 (+1.71)     |
> ```
>
> (We discuss the anomalous drop of GPT-5-chat on 3D Spatial Consistency in our response to R3 (Q1) and treat it as an outlier.)
>
> **Key observations.**
>
> 1. **Multi-view substantially reduces spatial ambiguity.**
>    The largest gains appear in *Distance Judgment*, which is most sensitive to depth and occlusion: GPT-5 and GPT-4.1 gain **+18.90** and **+10.44** points, respectively. This shows that many instances are genuinely under-constrained from a single image, while additional views make relative distances and occlusions resolvable.
>
> 2. **Leveraging multi-view input itself requires strong reasoning.**
>    Stronger models (GPT-5, GPT-4.1) **consistently benefit** from multi-view input overall and on the more spatially demanding subtasks. Some smaller models (e.g., Qwen2.5-VL-32B) show **small or even negative** Δ on certain subtasks, indicating that simply adding views can *hurt* when the model cannot robustly fuse partially misaligned observations. Thus our benchmark does not merely reward larger visual context; it stresses **non-trivial multi-view spatial reasoning**.
>
> 3. **A large gap to human multi-view performance remains.**
>    Even with multi-view input, models are far below human accuracy (≈91% on these subtasks). More views alone are not enough; current VLMs still lack reliable embodied spatial understanding in multi-view robotic scenes.
>
> Overall, these experiments directly address the concern about the necessity of multi-view inputs: (i) distance- and depth-sensitive tasks are markedly harder in the single-view setting, (ii) the benefit of multi-view input emerges mainly in stronger models that can integrate multiple viewpoints, and (iii) there remains substantial headroom to human performance, underscoring the value of MV-RoboBench as a challenging multi-view benchmark.
>
> Best regards,
> The Authors

---

> ### Author Response · Authors · 2025-11-22
> **Response to Reviewer oGrs – Weakness 2**
>
> Dear Reviewer,
>
> We thank the reviewer for this helpful clarification and agree that the phrase **“robotic task execution”** can be misleading if interpreted as **low-level motor control**. Our benchmark is intended to evaluate the **high-level reasoning** that precedes execution, not the full end-to-end control stack.
>
> In modern VLA systems, the vision–language component primarily performs **multi-view perception**, **affordance understanding**, and **high-level planning**, while separate policy modules handle low-level control. **MV-RoboBench is explicitly targeted at this reasoning layer.** To avoid overstating our scope, we have revised the manuscript to use more precise terminology such as **“robotic task reasoning”** and **“reasoning for high-level planning”** instead of “robotic task execution” in general. For example, in the abstract we now state that **“spatial intelligence and robotic task reasoning are correlated in multi-view robotic scenarios.”**
>
> We agree that directly fine-tuning VLAs for **low-level control on real robots** is an important direction, but it introduces additional confounding factors beyond spatial reasoning. We therefore focus this work on a **clean, diagnostic evaluation of the high-level reasoning component**, and view controlled VLA execution studies as promising future work. A more detailed discussion of the role of VLA-centric evaluation is provided in our response to **Reviewer Awkn (Awkn, Weakness 3)**.
>
> Best regards,
> The Authors

---

### Author Response · Authors · 2025-12-01
**Author Summary for Area Chair**

Dear Area Chair,

Given the unusual circumstances surrounding the OpenReview incident and the frozen discussion period, we provide a concise summary of the post-discussion status, which we hope will be helpful for your final assessment.

---

#### **Initial Reviews**

The paper received positive evaluations from all reviewers, with scores of **8, 6, 6, and 6**. Reviewers consistently highlighted three strengths:

* **Contribution:** A benchmark addressing an open evaluation gap in **multi-view spatial reasoning for robotic manipulation**.
* **Rigor:** Human-validated annotation, synchronized multi-view design, ablations, robustness testing, and evaluations across diverse model families.
* **Clarity & relevance:** Clear framing and direct applicability to VLM/VLA research.

---

#### **Rebuttal and Updates**

During the rebuttal phase, we focused on the key concerns raised by reviewers and introduced targeted updates:

* **Multi-view necessity:** Added single-view vs. multi-view ablations showing substantial improvements (e.g., **+18.9 points** for GPT-5 on depth-sensitive tasks).
* **Benchmark positioning:** Added explicit comparison with ERQA and MMSI-Bench to clarify scope and novelty.
* **Terminology clarification:** Updated language to emphasize **high-level robotic reasoning**, aligned with reviewer feedback.
* **Diagnostic completeness:** Added robustness experiments (orientation sensitivity and rejection-case evaluation) and an error analysis.

These updates were intended to directly address the concerns raised and improve clarity and completeness.

---

#### **Discussion Outcome**

Following these revisions, two reviewers indicated that their concerns had been fully resolved:

* **Reviewer Awkn (final score: 8):** *“The clarifications make the contribution clear. I will keep the score of 8.”*
* **Reviewer bybw:** *“The updates address my concerns.”*

Additionally, **Reviewer drdc noted in the initial review** that their score could increase if clarifications were made; the rebuttal and added analyses were targeted at these specific points. No reviewer raised remaining objections after the rebuttal.

---

#### **Current Evaluation Context**

Based on the discussion and the acknowledgements from reviewers, our understanding is that the main concerns have been addressed and the evaluation has moved in a positive direction, though we are happy to provide any further clarifications if requested.

---

Thank you for taking the time to evaluate our work despite the unusual discussion circumstances. We sincerely appreciate your work and consideration.

Sincerely,
The Authors

---

### Meta-Review · Area_Chair_zNrK · 2026-01-07

**Summary:**

This work proposes a benchmark, MV-RoboBench, for spatial reasoning of VLMs in robotics scenes. It receives 1 clear accept and 3 borderline accepts before rebuttal. Reviewers recognized the contributions, rigor, and clarity of this work, but raised concerns about the necessity of multi-view, benchmark positioning, terminology clarification, and diagnostic completeness.

**Reviewer Concerns:**

Reviewer concerns are well addressed during the rebuttal. Two reviewers indicated that they are satisfied with the rebuttal.

**Reviewer Scores:**

The reviewer questions are addressed during the rebuttal, and two reviewers commented that they are satisfied with the rebuttal. It is likely that the other two reviewers will either keep their borderline positive score or increase their scores to more positive ones.

---

### Decision · Program_Chairs · 2026-01-26

Accept (Poster)